

# On-line Differentiation of mineral phase in aerosol particles by Ion Formation mechanism using a LAAP-ToF single particle mass spectrometer

Nicholas A. Marsden[1], Michael J. Flynn[1], James D. Allan[1,2], Hugh Coe[1]

[1]School of Earth and Environmental Science, University of Manchester, Manchester, M13 9PL , UK
[2]National Centre for Atmospheric Science, Manchester, M13 9PL, UK

*Correspondence to*: Hugh Coe (hugh.coe@manchester.ac.uk

**Abstract.** Mineralogy of silicate mineral dust has a strong influence on climate and eco-systems due to variation in physicochemical properties that result from differences in composition and crystal structure (mineral phase). Traditional off-line methods of analysing mineral phase are labour intensive and the temporal resolution of the data is much longer than many atmospheric processes. Single particle mass spectrometry (SPMS) is an established technique for the on-line size resolved measurement of particle composition by laser desorption ionisation (LDI) followed by time-of-flight mass spectrometry (TOF-MS). Although non-quantitative, the technique is able to identify the presence of silicate minerals in airborne dust particles from markers of alkali metals and silicate molecular ions in the mass spectra. However, the differentiation of mineral phase in silicate particles by traditional mass spectral peak area measurements is not possible. This is because instrument function and matrix effects in the ionisation process result in variations in instrument response that are greater than the differences in composition between common mineral phases.

In this study, a novel technique is introduced that enables the differentiation of mineral phase in silicate mineral particles by ion formation mechanism measured from subtle changes in ion arrival times at the TOF-MS detector. Using a combination of peak area and peak centroid measurements, we show that the arrangement of the interstitial alkali metals in the crystal structure, an important property in silicate mineralogy, influences the ion arrival times of elemental and molecular ion species in the negative ion mass spectra. A classification scheme is presented that allows for the differentiation of illite/smectite, kaolinite and feldspar minerals on a single particle basis. On-line analysis of mineral dust aerosol generated from clay mineral standards produced mineral fractions that are in agreement with bulk measurements reported by traditional XRD analysis.

## 1 Introduction

Aerosol has a strong environmental impact by affecting climate, atmospheric processes, ecosystems, human health and visibility. Airborne mineral dust, which accounts for a large fraction of the global aerosol burden (Cakmur et al., 2006), influences climate by the direct radiative perturbation (Balkanski et al., 2007; Nousiainen et al., 2009; Tegen and Lacis,




1996), affecting cloud properties (DeMott, 2003; Rosenfeld et al., 2001) and atmospheric chemistry (Usher et al., 2003). In addition, mineral dust provides nutrients to land and ocean (Duce and Tindale, 1991; Jickells and Spokes, 2001) , causes damage to human health through the inhalation of fine particulate matter (Prospero et al., 2008; Prospero and Mayol-Bracero, 2013; Samoli et al., 2011) and causes disruption to transport and economy with intense episodes of reduced

visibility (Goudie and Middleton, 2006; Prospero, 1999).

Ambient measurement of mineral dust composition are important for validation of global dust cycle models and their incorporation into atmospheric models (Claquin et al., 1999; Nickovic et al., 2012; Perlwitz et al., 2015b). Differentiation of mineral phase is particularly useful for the provenance of transported dust because distinct mass fraction ratios of mineral types, such as calcite content and illite/kaolinite ratio, can be used as signatures of potential source area (PSA) (Caquineau et

al., 2002; Scheuvens et al., 2013). However, the relative fraction of minerals evolves during transport, so that the mineralogical distribution close to source is different to that after long range transport. In general, measurements made within or close to PSAs show a relatively high proportion of large grains (> 20 µm) of quartz and feldspar that become depleted by gravitational settling and sorting during transport so that mineral dust populations measured far from PSAs are dominated by the clay fraction (Perlwitz et al., 2015a). The source variability and sorting during transport have important

consequences for atmospheric processes and for fertilisation of the ocean such with phosphate and soluble $Fe^{2+}$ (Journet et al., 2008).

The role of a mineral dust particle in the atmospheric processes is a function of its physical and chemical properties which can be influenced by source and transport processes. Composition, external and internal mixing state, absolute aerosol mass concentration, particle size distribution and morphology/shape are important properties in many atmospheric processes

(Formenti et al., 2011). Recently, the importance of chemical and mineralogical composition has been highlighted. Optical modelling of mineral dust aerosol has found that the radiative effects are very sensitive to internal and external mixing of mineral types, with iron oxide – clay mineral aggregates of particular importance (Lafon et al., 2006; Sokolik and Toon, 1999). The effective parameterisation of radiative transfer models requires the accurate representation of internal and external mineral combinations derived from laboratory and field measurements. For example, McConnell et al. (2010 found

that radiative closure experiments required accurate representation of aggregates of illite or kaolinite clay minerals with hematite or goethite (iron oxide minerals), but the mineralogical combination had to be assumed due to the inability of traditional X-ray diffraction techniques to accurately resolve the mineralogy on sparsely populated filter samples collected during airborne measurements.

Radiative forcing due to change in cloud properties continues to be the biggest source of uncertainty in climate change

prediction (Seinfeld et al., 2016), with the availability of ice nucleating particles (INP) in cold and mixed phase clouds a large source of this uncertainty. Current climate models rely on simple parameterisation of INP number concentrations using properties such as surface area (Niemand et al., 2012) or particle size (DeMott et al., 2010, 2015), which assume all mineral dust behave in the same way regardless of chemical or mineralogical composition. However, laboratory experiments have found a strong mineralogical dependence on the onset temperature of ice nucleation with respect to supersaturation, which





have implications for modelling of cold and mixed phase clouds (Zimmerman 2008; Connolly 2009). For example, Atkinson et al. (2013) demonstrated that from a collection of common minerals, K-feldspar particles were the most efficient INP in the immersion mode using cold stage apparatus.

Heterogeneous ice nucleation in the natural environment can occur by multiple mechanisms that are difficult to recreate in the laboratory. Ice nucleation with particles in suspension using a variety of chamber techniques such as continuous flow diffusion chamber (CFDC), and controlled expansion cloud simulation chamber (CECC) create a realistic cold or mixed phase cloud simulation, but create difficulties in observing and measuring the actual INP in ice crystals as they must be extracted from the chamber as ice residue. In a comprehensive comparison of IN measurement techniques, Hiranuma et al., 2015 commented that differences in the IN efficiency of illite NX powder reported by a variety of techniques were not only a result of instrument function, but also differences in the physiochemical properties of the particles due to particle dispersion techniques and batch differences in the mineralogical composition of the sample. Representative samples of mineral dust aerosol for laboratory studies are difficult to source, and even nominally pure mineral samples can be heterogeneous in composition and contain substantial impurities that could be a significant source of INP. The ability to measure the mineralogy of a dust particle in real time would offer a huge advantage in the evaluation of both the starting material and the INP fraction in these types of studies.

In the field of atmospheric science, mineral dust is often collected in small volumes on microporous filters which are then analysed off-line by a variety of elemental, mineralogical and isotopic techniques. Typical techniques for elemental analysis include environmental scanning electron microscopy (ESEM), which reveals information on particle morphology, coupled with energy dispersive X-ray spectroscopy (EDS) e.g. (Reid et al., 2003; Young et al., 2016) or transition electron microscopy (TEM) e.g. (Kandler et al., 2007) for elemental analysis on a particle by particle basis. While elemental composition is strongly correlated to mineralogical composition, the accuracy of quantitation required to differentiate common silicate mineral phase distribution in the bulk sample is rarely achieved. X-ray diffraction (XRD) is the established technique used to identify major mineral phases in many disciplines. However, this approach is considered semi-quantitative and is subject to comparatively large errors, particularly when dealing with clay minerals in the fine fraction (< 2 µm) (Moore and Reynolds, 1997), that require the preparation of textured samples. In addition, the XRD signal results only from the crystalline fraction so that the amorphous material is not counted leading to a discrepancy with the total dust mass (Formenti et al., 2008). The limit of detection of XRD analysis mean a mass loading of at least 800µg of aerosol dust are typically required (Caquineau et al., 1997), making it unsuitable for the measurement of mineral dust particles with low number concentrations.

On-line measurements of aerosol properties are highly desirable becuase changes in properties can be linked to atmospheric processes and artefacts associated with off-line particle collection can be avoided. Single particle mass spectrometry (SPMS) is an on-line measurement technique that obtains size resolved composition of individual particles using laser desorption ionisation (LDI). In SPMS, aerosol is directly introduced into the instrument via an inlet and a high powered laser is used to ablate and ionise refractory and non-refractory particles in a particle beam. Ions produced in a low



vacuum source by the LDI process are analysed by time-of-flight mass spectrometry (TOFMS). The single particle nature of the technique is an advantage over bulk measurement because composition and mixing state of distinct particles can be obtained and directly correlated with other physiochemical properties such as particle size and density if used in tandem with other measurement techniques. However, the composition measurement is considered non-quantitative on a single particle
basis due to variation in instrument function and particle matrix effects that influence the ion formation process (Hinz and Spengler, 2007; Murphy, 2007; Zhou et al., 2007).

Numerous studies of atmospheric aerosol using SPMS have identified a distinct silicate mineral class of particles in ambient aerosol  (Dall'Osto et al., 2010; Middlebrook, 2003; Sullivan et al., 2007), and in ice residues extracted from mixed phase clouds (Baustian et al., 2012; Kamphus et al., 2010; Schmidt et al., 2016; Worringen et al., 2015).  Whilst this has
proven useful in the investigation of internal and external mixing states of ambient aerosol population, identification of mineral types within the silicate class has remained elusive.  Natural variation in composition between mineral phases is often smaller than the particle to particle variations in ion distribution recorded in the mass spectra due to instrument function and particle matrix effects. The origin of this strong matrix effect is not well understood, but systematic variation of ion signals in simple analogues of atmospheric aerosol suggest that ionisation potential, electron affinity and plume density
are important factors (Reinard and Johnston, 2008).

Ion formation processes in LDI can be studied by measuring initial ion velocities which are calculated by comparing the ion arrival time at the TOFMS detector of certain ion species after systematic variation in ion focussing (e.g Spengler & Kirsch 2003; Vera et al. 2005). In this paper we compare ion arrival times of key markers ions in a variety of crystal structures in otherwise chemically similar particle types. Using a Laser Ablation Aerosol Particle Time-of-Flight (LAAP-
TOF) single particle mass spectrometer (Aeromegt GmbH), we demonstrate that systematic variation in ion arrival times in nominally pure mineral samples can be related to the ion formation mechanism and crystal structure of single particles. A method for the on-line differentiation of mineral phase in clay mineral standards is presented using spectral peak centroid as a measure of average ion arrival time in addition to traditional peak area analysis.

## 2 Methods

### 2.1 Dust Samples

Laboratory studies of the physiochemical properties of mineral dust require an appropriate sample material that is representative and relevant to atmospheric processes. Field studies show that most dust in the atmosphere is dominated by varying quantities of quartz, k-feldspar, plagioclase, calcite, hematite, kaolinite, and the illite/smectite group of clay minerals (Formenti et al., 2008; Jeong, 2008; Kandler et al., 2007, 2009, 2011; Kaufman et al., 2005; Kaufmann et al., 2016). At a
single particle level, particle size dependence is observed with an abundance of quartz and feldspar grains in the coarse fraction and a fine fraction (< 2 μm) dominated by clay minerals. Detailed studies of the internal structures of Asian and Saharan dust have revealed complex internal structures of individual particles in terms of mineralogy and morphology. TEM



analysis of sliced particles has revealed that the most common sub-5 μm particle type was clay-rich agglomerate, dominated by nano-thin platelets of illite–smectite series clay minerals (ISCM) with submicron grains of iron (hydr)oxides (goethite and hematite) commonly dispersed through the particles (Jeong et al., 2016; Jeong and Nousiainen, 2014).

In our laboratory study of silicate dust, clay mineral samples were chosen because they represent the most atmospherically relevant material in the size range of the LAAP-TOF transmission (0.5-2.5 μm). Samples of feldspar were compared to these clay samples because of the potential importance of K-feldspar as ice nuclei (Atkinson et al., 2013). In addition to aluminosilicates, 2.1 μm borosilicate glass spheres were used as an example of an amorphous silicate structure that contains abundant alkali and earth alkali metals that are not chemically bonded into a crystal structure. Carbon black (CB) (Elftex 124, Cabot Corp) was used as an example of a non-silicate particle that has a well characterised molecular structure. A summary of the samples used in this study is provided in **Table 1**.

**Table 1. Summary of the dust samples used in this study**

| Sample Name | Principal Mineral | Structure Type | Sub-Type | Origin |
|---|---|---|---|---|
| BSG | Borosilicate Glass | Amorphous | N/A | Duke Standards |
| Elftex124 | Carbon Black | Sheet Graphite | N/A | Cabot |
| Ortho1 | Orthoclase | Framework Silicate | Feldspar | Geo Supplies |
| Plag1 | Plagioclase | Framework Silicate | Feldspar | Geo Supplies |
| KGa-1b | Kaolinite (low defect) | Sheet Silicate | 1:1 Layer Clay | CMS |
| KGa-2 | Kaolinite (high defect) | Sheet Silicate | 1:1 Layer Clay | CMS |
| STx-1b | Ca-Montmorillonite | Sheet Silicate | 2:1 Layer Clay | CMS |
| SWy-3 | Na-Montmorillonite | Sheet Silicate | 2:1 Layer Clay | CMS |
| IMt-2 | Illite | Sheet Silicate | 2:1 Layer Clay | CMS |
| ISCz-1 | Illite-Smectite Mix | Sheet Silicate | Mixed Layer Clay | CMS |
| Illite NX | Illite | Sheet Silicate | 2:1 Layer Clay | B+M Nottenkamper |

Carbon black (CB) is distinct from the material commonly referred to as black carbon (BC) (Long et al., 2013; Watson and Valberg, 2001). CB has a characteristic particle morphology that consists of spherical primary particles fused into aciniform (grape-like) aggregates which cluster into larger-sized agglomerates. The primary particles are typically 10-500 nm in diameter and are composed of imperfect graphitic layers that are concentrically arranged around a growth center (Rivin, 1986). The spaces between the graphitic layers often accommodate cations such as potassium and sodium. Elftex 124 is a flame soot derived CB product comprised of > 95 % carbon (Wonaschutz et al. 2009).

Clay minerals have a sheet silicate structure in which silicate tetrahedra are two dimensionally polymerised to form structural layers that are separated by interlayer cations. The capacity of a clay mineral to accommodate interstitial cations (X-ions) in the interlayer spacing of clay mineral is related to the layer charge of the structural layer. The structural layers of



clay minerals are composed of alternating tetrahedral and octahedral sheets in which the layer charge is created by isomorphous substitution of lower valency ions. Substitution of $Si^{4+}$ for $Al^{3+}$ ions in the tetrahedra produces tetrahedral charge while substitution of $Al^{3+}$ for $Mg^{2+}$ in the octohedra (Y-ions) produces octahedral charge. The composition of clay minerals is not fixed, but is determined by the varying degrees of cation replacement in the structural unit and interstitial complex. For example, the nominal chemical formula of illite is stated in the form $K_{1.5-1.0}Al_4[Si_{6.5-7.0}Al_{1.5-1.0}O_{20}](OH)_4$ to express the variability in composition that can exist within the illite member of the clay mineral group.

Source clays from the Clay Mineral Society (CMS) have been the subject of chemical, physical and thermodynamic analysis and therefore very well characterised. Average elemental composition of Si, Al, Fe(II), Fe(III), Mg, Ti, Mn, P, Ca, Na, K and $H_2O$ of the bulk sample is available (Mermut and Cano, 2001) and enables the construction of the precise structural formula. The structural formulae for the source clay samples used in this study are given in a format that shows the average composition of the X-ions, Y-ions and tetrahedral layer in **Table 2**.

Kaolinite (samples KGa-1b and KGa-2) has a 1:1 layer structure in which each layer contains 1 tetrahedral layer and 1 octahedral layer. Kaolinite has no overall layer charge and hence a very low X-ion content. The remaining samples consist of the so called 2:1 layer clays in which one octahedral layer is sandwiched between two tetrahedral layers. The layer charge created by this 2:1 layer structure is balanced by a certain quantity of X-ions. llite rich samples (IMT-2, ISCz-1) are characterised by very high quantities of $K^+$ whereas the montmorillonite clays (STx-1b, SWy-3) are characterised by relatively high levels of $Na^+$ and $Ca^{2+}$.

**Table 2. Structural formulae of source clays from the Clay Mineral Society calculated from elemental analysis** (Mermut and Cano, 2001)**.**

| Sample | X-ion | Y-ion | Tetra | Anion |
|---|---|---|---|---|
| KGa-1b | $Mg_{0.02}\,Ca_{0.01}Na_{0.01}\,K_{0.01}$ | $Al_{3.86}\,Fe(III)_{0.02}\,Ti_{0.11}\,Mn_{Tr}$ | $Si_{3.83}\,Al_{0.17}$ | $O_{10}(OH)_8$ |
| KGa-2 | $Ca_{Tr}\,K_{Tr}$ | $Al_{3.66}\,Fe(III)_{0.07}\,Ti_{0.16}\,Mn_{Tr}$ | $Si_{4.00}$ | $O_{10}(OH)_8$ |
| STx-1b | $Ca_{0.27}Na_{0.04}\,K_{0.01}$ | $Al_{2.41}\,Fe(III)_{0.09}\,Mg_{0.71}\,Ti_{0.03}\ \ Mn_{Tr}$ | $Si_{8.00}$ | $O_{20}(OH)_4$ |
| SWy-3 | $Ca_{0.12}Na_{0.32}\,K_{0.05}$ | $Al_{3.01}\,Fe(III)_{0.41}\,Mg_{0.54}\,Ti_{0.02}\,Mn_{0.01}$ | $Si_{7.98}\,Al_{0.02}$ | $O_{20}(OH)_4$ |
| IMt-2 | $Mg_{0.09}\,Ca_{0.06}\,K_{1.37}$ | $Al_{3.01}\,Fe(III)_{0.76}\,Fe(II)_{0.06}\,Mg_{0.43}\,Ti_{0.06}\,Mn_{Tr}$ | $Si_{7.08}\,Al_{0.92}$ | $O_{20}(OH)_4$ |
| ISCz-1 | $Mg_{0.03}\,Ca_{0.10}Na_{0.09}\,K_{0.95}$ | $Al_{3.39}\,Fe(III)_{0.12}\,Mg_{0.48}\,Ti_{Tr}\,Mn_{Tr}$ | $Si_{7.19}\,Al_{0.81}$ | $O_{20}(OH)_4$ |

Naturally occurring rock is rarely mono-mineralic. Mineralogical impurities in source clays have been previously quantified by X-ray diffraction analysis (**Table 3**) for the bulk sample (Chipera and Bish, 2001) and < 2 µm fraction (Vogt et al., 2002). For example, in natural environments montmorillonite is diagenetically altered to illite in the presence of K-feldspar (Garrels, 1984) so that clays often consist of fine interlayers of  these two clay minerals. Sample ISCz-1 is an example of clay that consists of microscopic interlayers of illite and smectite clay minerals (ISCM). Recently, Broadley et al. (2012) suggested illite NX (B+M Nottenkamper, Munich, Germany) as a suitable representation of ambient mineral dust





sampled at remote locations and it has been used in numerous ice nucleation studies (Hiranuma et al., 2015). Illite NX is a clay rich nanopowder that contains significant mineralogical impurities. XRD analysis has shown significant variation in the impurity content that may represent significant batch differences in mineralogy (**Table 4**).

**Table 3. Minerological impurities in CMS clay as determined by X-ray diffraction.**

| Sample | Impurities (wt%) | | | | | | |
|--------|--------|-----------|-----------|-------------|------------|-------|-----------|
|        | Quartz | Kaolinite | K-Feldspar | Plagioclase | Other | Total | Reference |
| KGa_1b |        |           |           |             | Dickite 4.0 | 4 | Chipera 2001 |
| KGa-2  |        |           |           |             | Dickite 4.0 | 4 | Chipera 2001 |
| PF1-1  | 1.7    | 1.7       | 5.8       | 2.5         |            | 11.7 | Vogt 2002 |
| STx-1b | 30     | 2.1       | 0.9       | 0.6         |            | 33.6 | Vogt 2002 |
| SWy-3  | 5.2    | 1.3       | ND        | 0.1         | Pyroxene 5.8 | 12.4 | Vogt 2002 |
| IMt-2  | 3.3    | 0.7       | 4.6       | 1.5         |            | 10.1 | Vogt 2002 |
| ISCz-1 | 0.7    | 0.2       | 2.3       | 5.5         | Chlorite 1.2 | 8.7 | Vogt 2002 |

**Table 4. Minerological impurities in illite NX as determined by X-ray diffraction.**

| Study | Impurities (wt%) | | | | | |
|-------|--------|-----------|-----------|-------------|-------|-------|
|       | Quartz | Kaolinite | K-Feldspar | Plagioclase | Other | Total |
| Manufacturer | 4 | 10 | ND | ND | ND | 14 |
| Hiranuma 2015 | 3 | 10 | 14 | ND | Calcite 3.0 | 30 |
| Broadley *et al* 2012 | 7 | 7 | 10 | ND |  | 24 |
| Friedrich et al 2008 | <1 | 5 | 4 | 1.1 | Phlogopite 7.8, Anhydrite 1.4 | 20 |

In framework silicates such as feldspar, the silicate structure is formed from 3 dimensionally polymerised silicate tetrahedral. In contrast to the interlayer cation structure of clay minerals, a low valence cation is held in an interstitial cavity
10   that balances the charge deficit created by substitution of $Al^{3+}$ for $Si^{4+}$ in the tetrahedral. Unlike clay minerals, the interstitial cations are fixed and not exchangeable during normal diagenetic processes. Most naturally occurring feldspars are not homogenous but contain separate potassium rich and sodium rich phases. Specimens of orthoclase (K-Feldspar) and plagioclase were purchased from Geo Supplies Ltd, and analysed by XRF analysis to determine the average elemental composition. The average structural formula of the samples was then calculated from the relative proportions of the metal
15   oxides (Table 5).

*Table 5. Structural formulae of feldspar sample calculated from XRF analysis.*

| Sample | Interstitial X-ion | Tetra | Anion |
|--------|-------------------|-------|-------|
| Orthoclase | $Ca_{0.01}Na_{0.25}K_{0.85}$ | $Si_{2.95}Al_{1.02}$ | $O_8$ |



| Plagioclase | $Ca_{0.16}Na_{0.76}K_{0.07}$ | $Si_{2.83}Al_{1.17}$ | $O_8$ |
|---|---|---|---|

## 2.2 Experimental Setup

Mineral dust aerosol was sampled by the LAAP-TOF after dry dispersion of powder in a home-made dust tower and transmission of the extracted aerosol through a centrifugal particle mass analyser (CPMA) (Cambustion, Ltd) (Olfert and

Collings, 2005). Source clays, illite NX, borosilicate glass beads and CB were supplied in powder form or loose aggregate that was easily broken down into powder by gentle abrasion with a pestle and mortar. The feldspar samples were supplied as a large crystal that had to be reduced to powder in a timor mill prior to injection into the dust tower.

Powdered mineral dust was loaded into a modified sample vial and injected into the dust tower inlet using a short burst of compressed air from an air duster can. The mixing and suspension of mineral dust particles was achieved by turbulent flow,

controlled with an adjustable flow of compressed air introduced into the bottom of the tower (Figure 1). Aerosol was drawn out of the dust tower through the CPMA using a pumped sampling line throttled to a suitable flow rate (~1.6 L/min) using a needle valve. The flow was divided between the pumped line and the LAAP-TOF inlet (sampling at 0.078 L/min) using a `Y piece'.

The experiment was operated in two modes. In polydisperse mode, the CPMA was inactive so that the mass distribution

of the aerosol analysed by the LAAP-TOF was determined by the transmission efficiency of the aerodynamic lens.  In monodisperse mode, the CPMA was used in static mode so that only the selected particle mass was transferred to the LAAP-TOF.

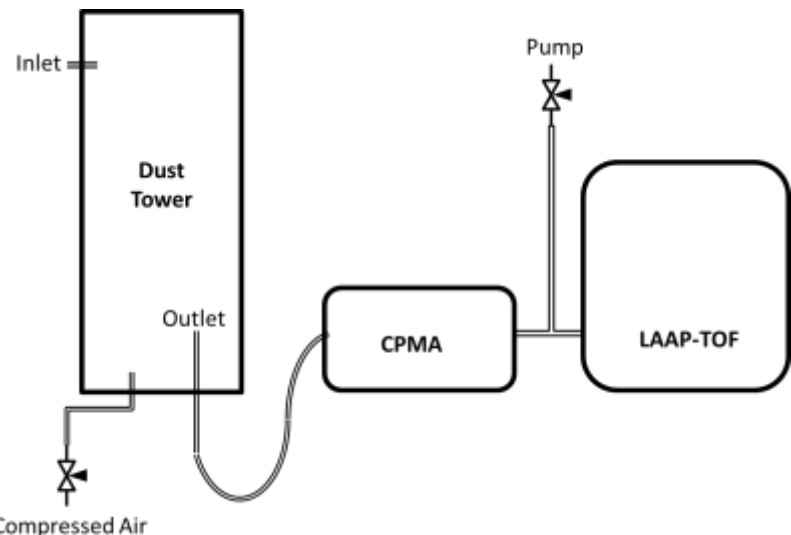

*Figure 1.  Schematic representation of the experimental setup.*





### 2.3 LAAP-ToF Single particle mass spectrometer

The LAAP-TOF used in this study is a modified version of the commercially available single particle mass spectrometer manufactured by AeroMegt (GmbH) that features a modified optical detection system as described in detail by Marsden et al. (2016). Briefly, particle laden air enters the instrument via an aerodynamic lens inlet (Liu et al., 1995) which produces a

narrow, but divergent particle beam along the instrument axis.  A pulsed excimer laser interacts with a particle in the low vacuum source region towards the back of the instrument (Figure 2). An optical particle detection stage, located within the source region, is used for temporal alignment of the excimer laser pulse with the presence of a particle in the ion source. The option to measure aerodynamic particle size using an additional optical particle detection stage located upstream on the instrument axis was not used in this study.

The version of the instrument used in this study features an aerodynamic lens inlet (model LP2.5 Aeromegt GmbH) for transmission of particles approximately 0.07-2.5 µm in diameter that is similar in design to the high pressure lens (HPL) described by Williams et al. (2013). Divergence of the particle beam is size and shape dependent (Jayne et al., 2000), resulting in a morphological dependence on the fraction of particles that reach the ionisation region (Huffman et al., 2005), an effect that is particularly important for measurements of irregularly shaped platy mineral in polydisperse dust.

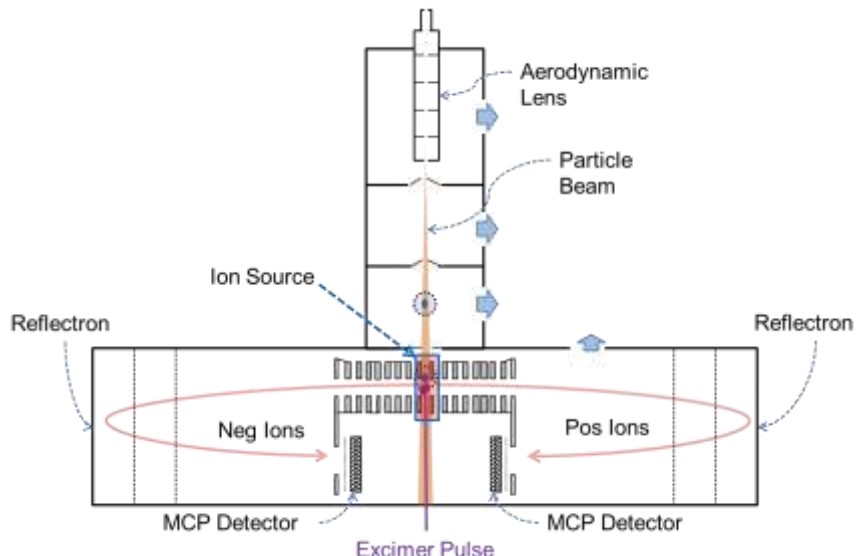

*Figure 2.  Schematic of the LAAP-TOF*

Laser desorption ionisation takes place in a high vacuum (<1e-6 mbar ) region some 230 mm from the aerodynamic lens exit, where particles interact with a 8 ns pulse of 193 nm radiation that is fired co-axial but counter-propagate to the particle beam. The ablation laser, an ArF excimer (model EX5, GAM Laser Inc.), can be requested to produce 2-12 mJ per pulse via

a setting in the software. The peak laser power actually encountered by a particle is unknown as the particle will transect the laser pulse at an unknown position in its Gaussian profile due to particle divergence and temporal alignment. Consequently,




particle to particle variations in the peak power and effective pulse duration can reduce the reproducibility of the measurement. A previous study with this instrument configuration has shown a sampling efficiency for ambient mineral dust of approximately 0.01 with a particle size dependent optical detection bias that favours the detection of particles with a mode of around 1.5 μm in diameter (Marsden et al., 2016). Sampling efficiency is important in SPMS as it influences the particle

counting statistics of an aerosol population and can introduce a bias of compositional measurement towards a particular particle type (Zelenyuk et al., 2009).

The TOFMS used in the LAAP-TOF is a bipolar reflectron TOF analyser (BTOF, Tofwerks AG) for the simultaneous measurement of positive and negative ions. Ion arrival times at the multichannel plate (MCP) detectors, one for each ion mode, are recorded by a dual channel 14bit analogue to digital converter (model ADQ214, SP Devices) with a bin width of

2.5 ns. The ADQ clock is triggered by a signal that is synchronous with the firing of the ablation laser so that each spectrum represents the ions formed by a single particle ablation event. The ion extraction conditions in the LAAP-TOF are unusual in that it utilises a field free extraction regime. The first extraction lenses of the LAAP-TOF ion optics are grounded so that the evolution of the ion plume takes place in a field free region and only ions with the correct ion trajectories are extracted for analysis.

**2.4 Mass spectral peak analysis**

The aim of the peak analysis is to evaluate how composition and crystal structure of the particles affect ion arrival times at the multi-channel plate (MCP) detector by comparing the spectral characteristics of the samples of mineral dusts of known mineralogy. TOF-MS analysers focus a certain elemental or molecular ion species into a discrete packet of ions whose arrival times at the detector are represented by a Gaussian probability density function (PDF) which is represented as a

spectral peak in the data. The average ion arrival time (T) of the ion packet at the detector is described by the peak centroid while the peak width (FWHM) describes the distribution around this average.

Peak centroid and peak width were extracted from the raw spectra using a peak fitting algorithm. Peak fitting within the spectra was problematic because of spectrum to spectrum variations in peak position, peak shape and interference from neighbouring peaks. To reduce the influence of these effects, a multipeak fitting procedure (multipeak fit v2, igor v6.36) was

performed on a portion of the mass spectrum which contained the peak of interest. Peak fitting parameters were set so that interfering signals were identified as separate peaks without comprising the integrity of the peak of interest. In most cases, the peak of interest was a well-defined Gaussian shape that was sufficiently separated from its neighbour, such as the example of $SiO_3^-$ molecular ion (m/z -76) in Figure 3a. Occasionally, especially in illite rich clay mineral samples, the molecular ion peak was much broader (Figure 3b) therefore a number of data smooths were required to ensure satisfactory

peak definition was obtained. A third peak morphology, featuring multiple modes was also encountered in illite rich clays samples that required careful consideration. Using a suitable peak fitting parameters (Number of Smooths 5, Number Fraction 0.1), the multimodal peak in Figure 3c was resolved into two peaks whose peak centres were approximately 1 Da (50 ns) apart. Note that further resolution of this multimodal peak using a fewer data smooths resulted in peaks that were


separated by <0.5 Da (20ns) (Figure 3d), which is an improbable solution in data that primary consists of singly charged ions. Consequently, peak fitting parameters were chosen that ensured peak centroid spacing of integer values (in mass) were produced from all fitting. The origin of the multimodal peak morphology will be discussed in more detail in the results section.

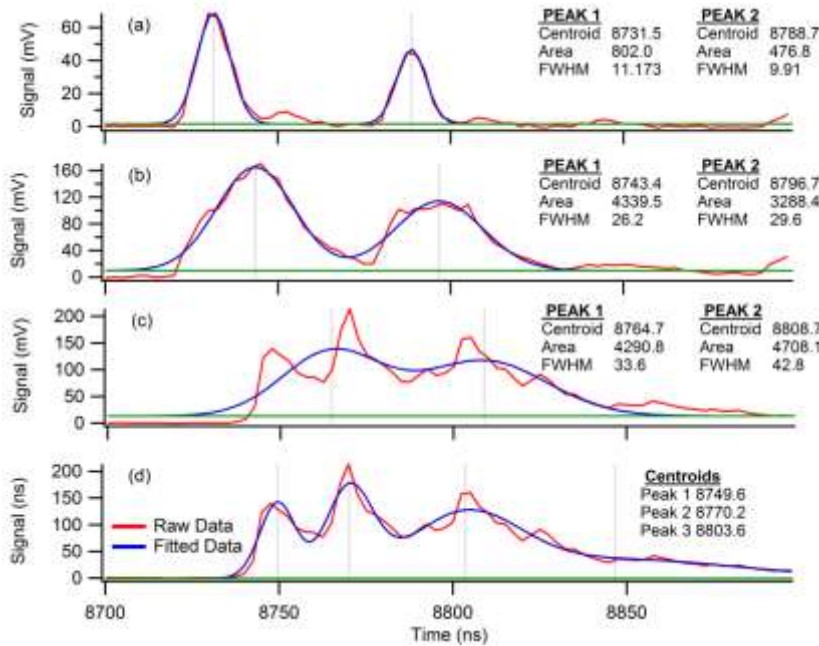

*Figure 3. Peak fitting of the SiO3⁻ molecular ion (m/z 76) and its neighbour at m/z 77. (a) Well resolved Gaussian peaks with a peak centroid spacing of 50ns, (b) peak broadening observed in illite rich material, (c) multimodal peaks resolved into two peaks with approximately 50ns spacing and (d) the same multimodal peak resolved into peaks with centroid spacing <50 ns by reducing the number of data smoothes to 2. Fitting parameters used were number of smoothes 5, Min Fraction 0.1 unless stated otherwise.*

Prior to analysing the mineral dust samples, the ion optics were tuned for maximum resolution and symmetrical peak shape across the mass range for positive and negative ions using 700nm diameter PSL particles. The effect of particle type on ion arrival times was initially evaluated by comparing mass scale calibrations obtained from selected samples. Calibration coefficients were calculated for each sample type by fitting the first order approximation of the time of flight equation (Eq. 1) to the averaged peak centroid of three ion species. For the silicate, the same ion species were chosen in positive and

negative ion modes for each sample. Changes in average arrival times of a certain ion species (i), apparent as a peak centroid shift ($\Delta T_{(i)}$) with respect to previously defined position, were examined in more detail by comparing the peak centroid with a nominal mass scale calibration and comparing $\Delta T_{(i)}$ with other ion species from within the spectrum (i.e. from the same ionisation event).

$T = a + b\sqrt{M}$                                                                                                      (1)





## 3 Results

### 3.1 Tuning and calibration

Spectral resolution of the peaks in positive and negative spectra for PSL were typically 300 and 600 (FWHM at m/z 36) respectively, with unit mass resolution easily achieved over the mass range 10-100 Da. The optimum tune settings for the positive ion optics differed somewhat from those of the negative ion optics despite their identical geometry, indicating significant differences in the translational energies of the positive and negative ions produced in the same particle ionisation event.

Typical mass spectra of feldspar mineral dust (**Figure 4**a) show elemental ions of alkali metals in positive ion mode and molecular fragment ions of silicate at m/z 60 ($SiO_2$) and m/z 76 ($SiO_3$) in negative ion mode which is in agreement with the mineral dust spectra observed by (Gallavardin et al. 2008a) who used a similar 193 nm excimer laser at approximately $10^9$ W/cm$^2$. Detector saturation was apparent on several positive ion peaks when Channel 1 signal amplifier line to the A/D was used (**Figure 4**a Top). To avoid peak saturation, the Channel 2 signal amplifier with a greater attenuation was used for the subsequent peak analysis. The effect of changing the signal amplifier on the spectra is demonstrated in in the typical mass spectra examples in **Figure 4**. Detector saturation occurs at 1200 mV signal strength for the peaks at m/z 27, 39 and 44 with the signal channel 1. Saturation is avoided with signal channel 2 but comes at the expense of poor signal to noise ratio of ion species such as $C^+$ (m/z 12) and $CaO^+$ (m/z 56). Note the relative signal height between ion species is altered when the signal channel is changed, highlighting insufficient dynamic range of the ion detection system.



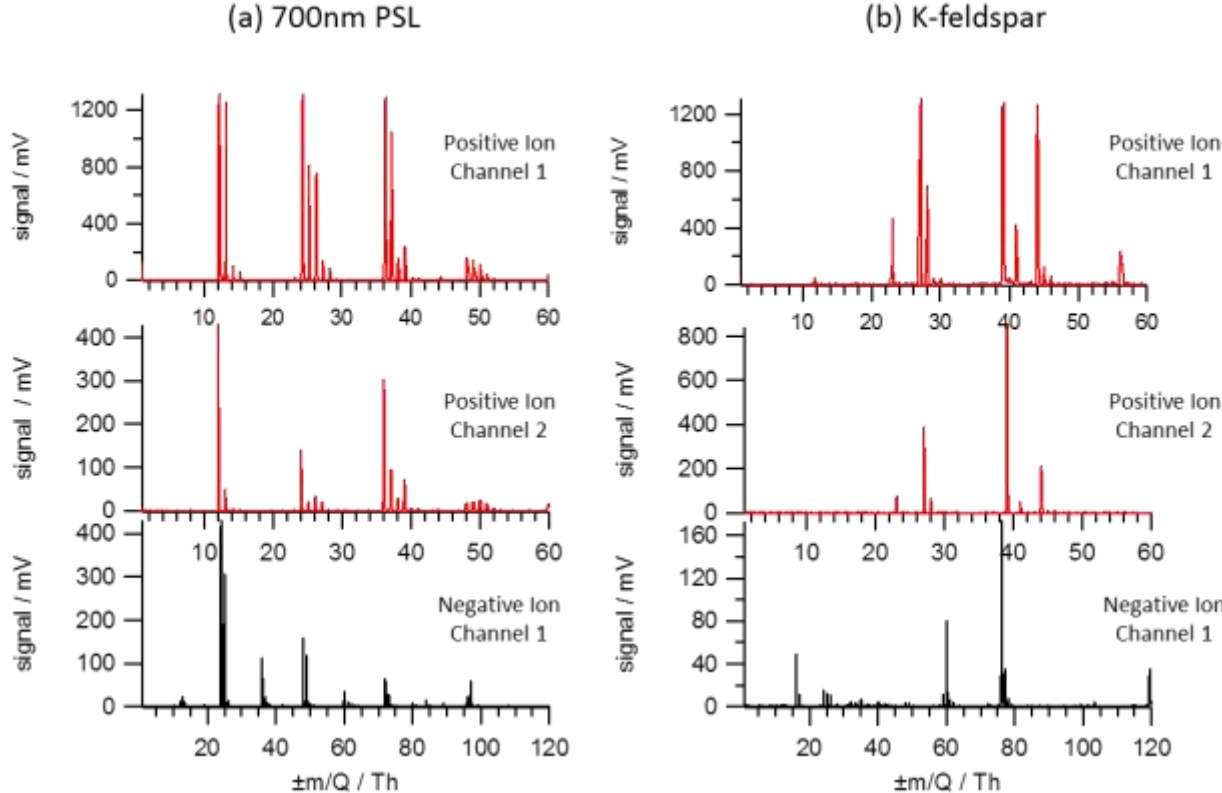

**Figure 4.** *Example spectra of (a) 700nm PSL and (left column) and (b) orthoclase feldspar (right column). The dynamic range of the data is demonstrated by changing the positive ion signal line from channel 1 (top spectra) to channel 2 (middle spectra). Negative ion spectra are also shown (bottom spectra).*

5  Mass scale calibration over a limited mass range (m/z 23-56) was possible in positive ion mode due to the limited number of peaks that were universally available in all spectra, whereas a mass scale calibration with a wider mass range (m/z 16-76) was possible in negative ion mode. Calibration coefficients obtained for PSL, Carbon Black, K-feldspar and borosilicate glass are shown in Table 6. Note the increase in the intercept value for the K-feldspar and borosilicate glass samples, indicating an overall shift in the ion arrival times compared to the PSL.

10  *Table 6. Mass calibration coefficients calculated from the mode peak position of 3 ion species in each ion mode. The ions species used are m/z +24,+ 36 +48, -24, -60 and -72 in PSL and Carbon Black mass spectra and m/z +27, +39, +44, -16, -60, and -76 in K-Feldspar mass spectra. Positive ion calibration of the borosilicate glass was not possible with the peaks selected. Calibration points are expressed in 2.5 ns bin widths.*

|  | *a* (Pos) | *b* (Pos) | *a* (Neg) | *B* (Neg) |
|---|---|---|---|---|
| **PSL** | 8.415 | 371.026 | 5.953 | 400.708 |
| **Carbon Black** | 16.1504 | 369.608 | 4.816 | 400.732 |
| **Feldspar** | 28.882 | 367.335 | 10.601 | 400.073 |
| **Borosilicate** | N/A | N/A | 34.125 | 397.435 |




The average peak position and peak width (± 2 SD) of the $Al^+$, $K^+$ and $SiO^+$ positive ion species and $O^-$, $SiO_2^-$ and $SiO_3^-$ negative ion species in the K-feldspar, Illite IMt-2 and borosilicate glass samples are displayed in Figure 5. In positive ion mode, the peak position is often > ±0.5 Da of the K-feldspar calibrated mass position in all sample types. In negative ion mode, the average position of the $SiO_2^-$ and $SiO_3^-$ is significantly larger for the Illite IMt-2 than for K-Feldspar or Borosilicate glass. In addition to the peak analysis, average peak centroid and peak width are important for the creation of stick spectra used in routine cluster analysis techniques. The data acquisition software provides an additional facility to perform an accurate calibration correction by shifting selected peaks to the corresponding unit mass position on the mass scale. However, for this calibration to be successful, the peak of interest must be located ±0.5 Da from the average mass position which is not the case in a significant number of peaks in mineral dust spectra.

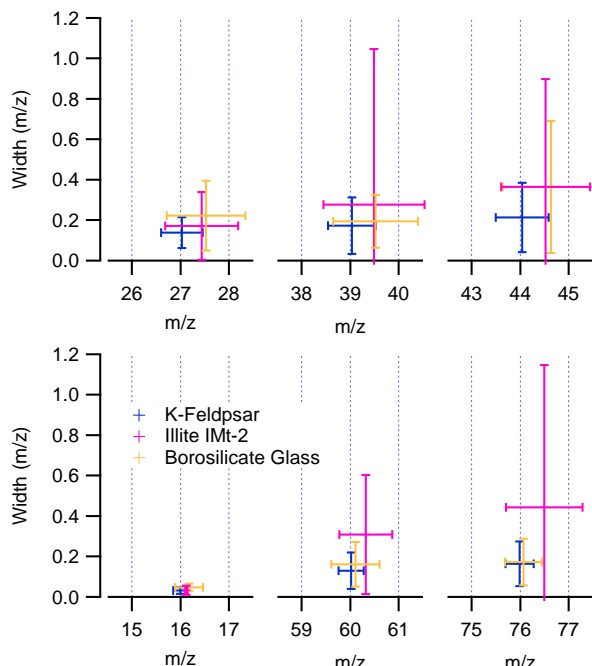

*Figure 5. The mean mass scale position and mass scale width of 3 positive ion species and 3 negative ion species in silicate mineral particles. Measurement is with respect to the K-feldspar calibration. Error bars are 2 standard deviations of the mean.*

The mass scale dependence of the peak position shift suggests that the differences in calibrations do not arise from a simple linear shift in ion arrival times for all ion species in the mass spectra. This is further demonstrated by examining $\Delta T_{(i)}$ with respect to the negative ion calibration for borosilicate glass (Figure 6). Carbon black has a clear mass dependence with respect to this calibration. Kaolinite does not have a strong mass dependence but is affected by a large kink in the mass scale between the $SiO_2$ and $SiO_3$ molecular ions (m/z 60 and 76 respectively), which is apparent with all the silicate minerals analysed.



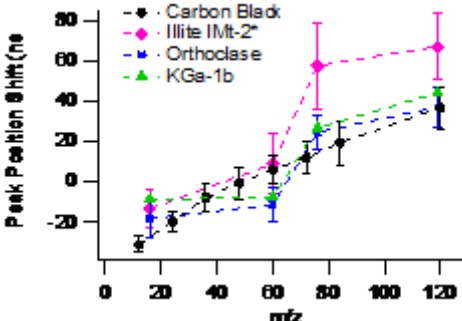

**Figure 6. Shift in ion arrival times ($\Delta T_{(i)}$) with respect to a nominal negative ion calibration with borosilicate glass. The mean and two standard deviations of the mean are plotted.**

### 3.2 Relative Shift of elemental and molecular ions

In this section we explore the relationship between the ion arrival times of O⁻ elemental ion ($T_O$) and the $SiO_3^-$ molecular ion ($T_{SiO_3}$) within discrete ionisation events. In the case of borosilicate glass, an amorphous silicate material, a scatter plot of arrival times measured in individual mass spectra display an approximately linear distribution of data points with a gradient of 1 (Figure 7a) indicating that $\Delta T_O \approx \Delta T_{SiO_3}$. In contrast, the non-silicate carbon black, a similar plot of elemental vs molecular carbon ions measured from C at m/z -12 and $C_6$ ion at m/z -60 respectively, gives a linear distribution of ion
arrival times with a gradient of 0.35, indicating that generally $\Delta T_C \approx 0.35 * \Delta T_{C_6}$.

Similar distributions of ion arrival times can be observed with the LDI of crystalline silicate mineral dust. The framework silicate orthoclase feldspar, has a distribution of ion arrival times that is similar to the borosilicate glass except the distribution is narrower (Figure 7b), whereas illite IMt-2, a sheet silicate, has a large mode that is similar to the carbon black, with an additional mode of particles that is more similar to the borosilicate glass. The silicate mineral sample with the most
variation in ion arrival times is the illite nx where several distinct modes are apparent (Figure 7c). The multimodal nature of this distribution was still apparent after the particles were mass selected with the CPMA before analysis with the LAAP-TOF.  The ion arrival time distribution of mass selected 350 fg illite NX particle shows a smaller variation of up to 20ns whereas mass selected 700 fg illite NX particles have a distribution comparable to the polydisperse analysis.

In order to directly compare the ion arrival time distributions of different silicate minerals, the shift in arrival time ($\Delta T_{(i)}$)
was calculated for each spectrum with respect to a set point on the time scale. For silicate containing particles, $\Delta T_O$ and $\Delta T_{SiO_3}$ were calculated with respect to the point where the mode distributions converge on the scatter plots at $T_O = 4010.2$ ns and $T_{SiO_3} = 8722.4$ ns respectively, which we will call the convergence point. The relative difference in the ion arrival times of the elemental and molecular ions in negative ion mode can then be expressed as a ratio ($\tau$) for each particle analysed.

$$\tau = \frac{\Delta T_O}{\Delta T_{SiO_3}}$$





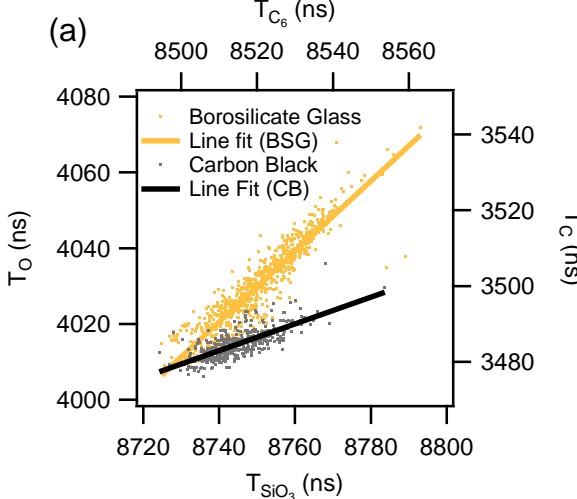

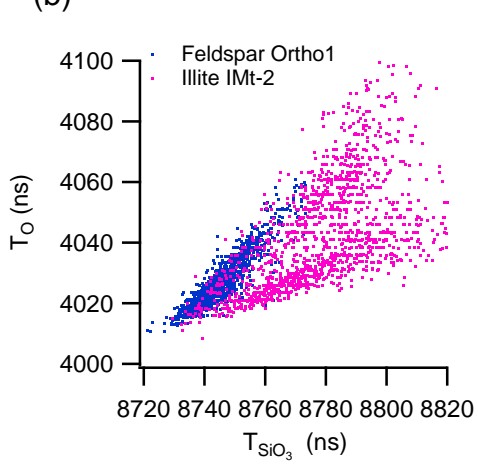

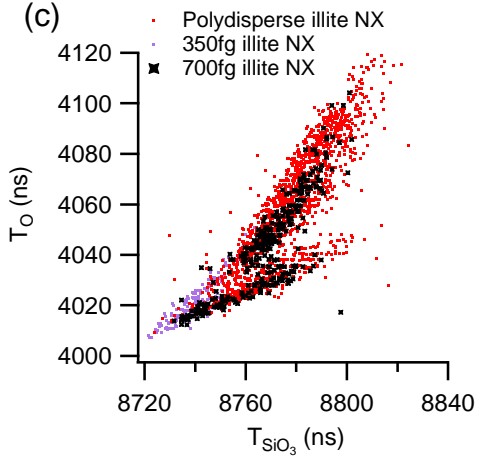




***Figure 7. Ion arrival times (T(i)) of elemental ions compared to molecular ion species in negative ion mode. Each point represents a single particle measurement. (a) 2.1µm borosilicate glass spheres and polydisperse carbon black. (b) polydisperse orthoclase feldspar and Illite IMt-2 and (c) polydisperse illite NX and mass selected illite nx particle with the CPMA.***

Histograms of τ values measured from the peak analysis in silicate mineral mass spectra are shown in Figure 8.

5 Potassium rich clay minerals illite NX and illite IMt-2 (Figure 8a) have a distinct mode around 0.3-0.5, that is aligned with

the $\Delta T_C/\Delta T_{C_6}$ ratio derived from the peak analysis of carbon black. Other distinct modes can be seen in the illite IMt-2

sample at 0.79 and in the illite NX sample at 0.83 and 1.10 that is more similar to the distribution of τ values measured for

borosilicate glass. The smectite and montmorillonite clays have modes in the distribution that range from 0.4 to 0.86 and

the 1:1 layer clay kaolinite has the largest mode at 0.93 (Figure 8b).

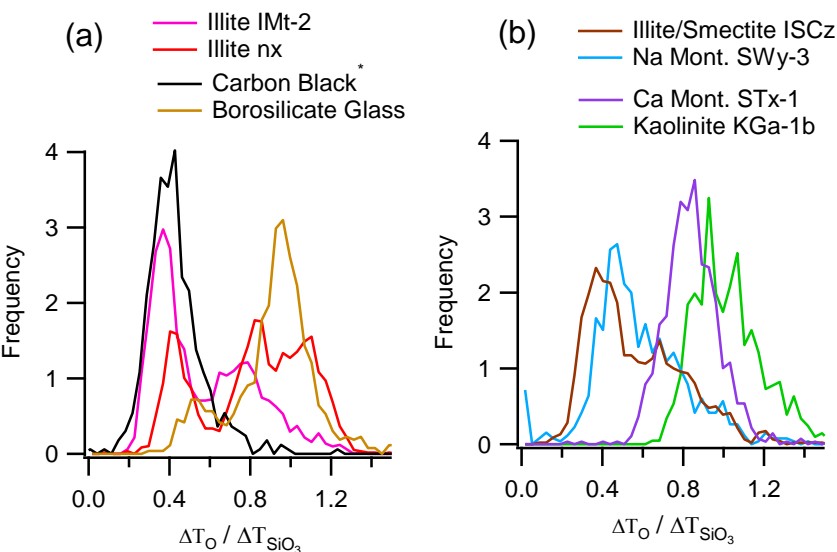

***Figure 8. Histograms of the ion arrival times shift ratio (τ) of the elemental ion O- and the molecular ion SiO₃- (a) potassium rich illite samples and (b) Na and Ca rich samples of montmorillonite and kaolinite. Borosilicate glass and carbon black are shown for reference. \*Carbon black calculated from the $\Delta T_C/\Delta T_{C_6}$ ratio.***

15 The main physiochemical difference between clay minerals is the nature of the interstitial X-ion that is accommodated

between the silicate structural layers. For the source clays, the relative proportion of low valence interstitial cations (X-ion)

present in the structure is discernible from the structural formulae provided by the clay mineral society (**Table 2**). These ion

proportions are listed alongside the literature values for interlayer charge and value of τ of the principal mode derived from

this study in **Table 7Error! Reference source not found.**. There is some correlation between τ and the layer charge, with the

20 exception of the calcium rich montmorillonite STx-1. In calcium rich montmorillonite, the layer charge is balanced by

abundant divalent ions of calcium, which has a first ionisation energy (589.8 kJ/mol), higher than the aluminium (577.5

kJ/mol) in the silicate structural layer. In contrast, SWy-1, ISCz-1 and IMt-2 samples have a layer charge that is principally




balanced by sodium and potassium ions that have first ionisation energies that are lower than any component in the silicate structural layer (495.8 and 418.8 kJ/mol respectively).

**Table 7.** *Measured values of τ compared to Interlayer charge and cation ratios from structural formulae derived from elemental analysis (Tr = Trace). Reference material from the clay mineral society.*

| Sample | InterLayer Charge | Interstitial Cation (X-ion) | | | Mode τ |
|---|---|---|---|---|---|
| | | Ca | Na | K | |
| KGa-1b | -0.06 | 0.01 | 0.01 | 0.01 | 0.93 |
| KGa-2 | 0.16 | Tr | 0 | Tr | 0.93 |
| STx-1 | -0.68 | 0.27 | 0.04 | 0.01 | 0.79 |
| SWy-2 | -0.55 | 0.12 | 0.32 | 0.05 | 0.47 |
| ISCz-1 | -1.29 | 0.10 | 0.09 | 0.95 | 0.37 |
| IMt-2 | -1.68 | 0.06 | 0 | 1.37 | 0.33 |

**3.3 The role of the interstitial complex**

A comparison of the interstitial X-ion content and the measured τ value for clay minerals and feldspars is given in *Figure 9*. The combined sodium and potassium content, derived from the calculated structural formulae for source clays is plotted against the mode value of τ derived from the peak analysis . For clay samples that exhibit multi-modal τ distributions, only the lowest principal mode is represented in the data. Relatively high levels of sodium and potassium result in a low τ value for these modes. For feldspars, the cation content is measured from the positive ion mass spectra and plotted against τ on a single particle basis for orthoclase and plagioclase in Figure 9a and Figure 9b respectively. In contrast, higher sodium and potassium content in feldspars result in high τ values. The influence of the calcium ion on the τ value in plagioclase feldspar is demonstrated in the colour function in plot Figure 9b. When Ca/Na ratios are high, the τ is >1 which is similar response as the kaolinite sample which is depleted in interstitial X-ions.

Mass selecting 700 fg orthoclase particles with the CPMA produces a similar distribution as polydisperse particles suggesting there is not particle size effect to the distribution. The low recorded potassium and sodium content in some feldspar particles may be a result of the crushing of the sample as this is known to cause collapse of the interstitial cavity (Garcia-guinea and Correcher, 2000) so that feldspar particles with low cation content may not be representative of natural occurring feldspar.





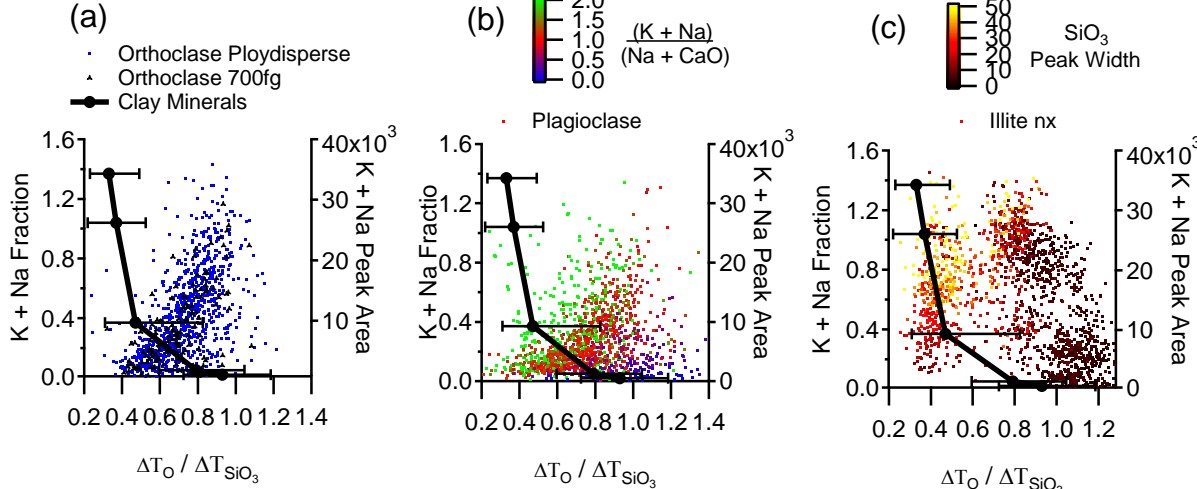

*Figure 9. Peak shift ratio (τ) with respect to the combined sodium and potassium content. Clay mineral are the mode value of R againt the cation content in the structural formula derived from elemental analysis. Error bars are the 25 and 75 percentile respectively. Feldspar cation content is measured from the positive ion mass spectral peak areas. (a) polydisperse and mass selected 700fg orthoclase fledspar particles and (b) plagioclase fledspar with the CaO content (m/z 56) highlighted.(c)The represntative clay mineral sample illite NX with the width of the SiO₃ molecular ion highlighted in the colour function.*

Distinctive clusters of data points are apparent in the illite NX sample when τ is plotted against alkali metal content derived from single particle mass spectra (Figure 9c). The concentration of particles in the region τ = 1.0-1.2 are clearly distinguishable from particles in the τ = 0.6-1.0 region by the content of low valence potassium and sodium ions. These clusters of data points occupy similar areas of the plot as kaolinite and feldspars respectively. The large cluster of particles with τ < 0.64 is congruent the principal mode in the Imt-2 and ISCz-1 samples, and is also characterised by relatively large peak width values. Multimodal peak shapes, as described in Figure 3d, mainly occur with τ > 0.64 in the illite NX sample but were never observed with feldspar where peak widths of the $SiO_3$ molecular ion were always < 15 ns. Broad (>15ns), multimodal spectral peak morphologies were attributed to aggregated or mixed particles in which the ion formation process is not uniform.

### 3.4 Single particle differentiation of clay mineral standards

The dependence of negative peak shift on the nature of the cations in the interstitial complex gives rise to the possibility of using these measurements to classify the mineralogical contents of mineral dust samples. Using published XRD analysis data as a guide, the single particle classification scheme outlined in**Error! Reference source not found.** Table 8 was used to quantify the particle number concentrations in the clay rich mineral samples illite NX, IMt-2 and ISCz-1 (Figure 10). The first class of particles is defined as the group of particles with τ < 0.64 which only appear in samples rich in illite clays. This class is attributed to illite-smectite clay minerals (ISCM) because in natural clay samples, illite occurs in fine interlayers with smectite clays. The second class, is particularly well developed in the illite NX and IMt-2 samples and is defined by τ > 0.64



and high potassium and sodium content that is typical of the orthoclase (K-feldspar) sample. The exception is particles within this group that exhibit broad multimodal peak morphology which are placed in a separate class of mixed particles (class 5). Kaolinite (class 3) has, by definition, very low potassium and sodium content and is also defined by high $\tau$ measured in the pure form of the mineral (KGa-1b above). The fourth mineral class assigns particles with intermediate values of $\tau$ and relatively low potassium and sodium content to montmorillonite or Na-Feldspar (plagioclase) because of the difficulty in distinguishing between these two minerals with the peak analysis. In cases where the peak analysis was unable to produce a result, mainly due to failure of the peak fitting, were allocated to a not classified particle class.

**Table 8. Single particle mineral phase classification scheme based on $\tau$, combined potassium and sodium content, and $SiO_3$ molecular ion peak width.**

| Class | Mineral Phase | $\Delta T_O / \Delta T_{SiO_3}$ ($\tau$) | K + Na Peak Area | $SiO_3$ Peak Width |
|---|---|---|---|---|
| 1 | ISCM | $< 0.64$ | | |
| 2 | K-Feldspars | $> 0.64$ | $> 15000$ | $< 15$ |
| 3 | Kaolinite | $> 0.885$ | $< 15000$ | |
| 4 | Montmorillonite/Plagioclase | $0.64 - 0.885$ | $< 15000$ | |
| 5 | Mixed | $> 0.64$ | | $> 15$ |
| 6 | Not Classified (NC) | | | |



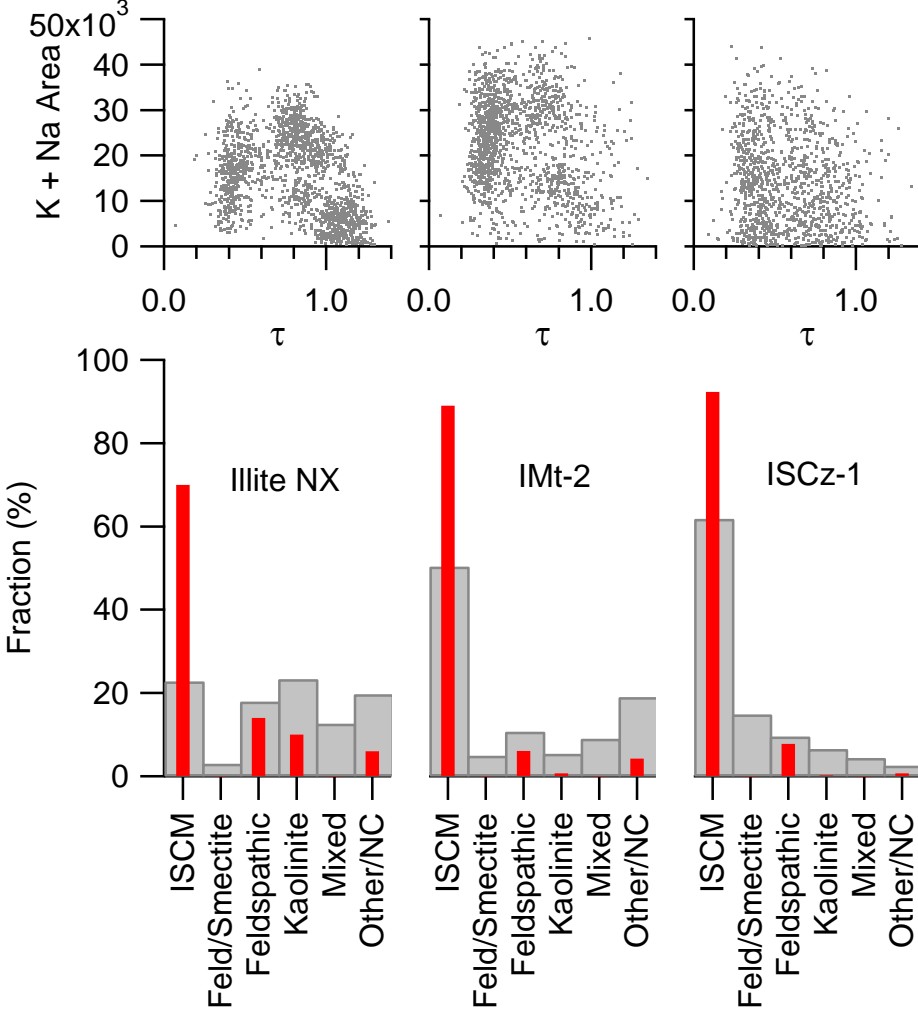

*Figure 10. Mineral phase classification of Illite NX, IMt-2 and ISCz-1 clay reference samples using a combined peak position shift and traditional peak area technique. Single particle measurements of $\Delta T_O / \Delta T_{SiO_3}$ (τ) vs the combined potassium and sodium content are shown in the top panel. The bottom panel report the particle number fractions determined by the peak analysis classification method. Mass fractions reported for XRD analysis of illite NX from Hiranuma et al.( 2015), and IMt-2 and ISCz-1 from Vogt (2002).*

The main difference in the mineral fraction reported by XRD analysis and this peak analysis technique is the relative fraction of ISCM minerals with non-ISCM minerals such as K-feldspar and kaolinite. The differences may arise from comparing a number counting technique with a mass fraction, and from hit-rate bias in the LAAP-TOF measurement which may discriminate against certain particle types due to morphology and composition. In addition, the XRD analysis technique does not report the unidentified fraction of such as amorphous glassy material that may contribute to the non-ISCM fraction in the peak analysis.



### 3.5 Relative shift of positive and negative ions

The ratio $\Delta T_K/\Delta T_O$ is plotted against $\Delta T_O/\Delta T_{SiO_3}$ ($\tau$), in **Figure 11**. In this plot, the distribution of data points for borosilicate glass and orthoclase feldspar suggest a positive correlation between these ratios, with the trends in the distributions converging towards a point where the magnitude of shift of all species is equal. In illite IMt-2 the ISCM mineral fraction, identified as class 1 in the classification scheme, does not show a correlation between positive alkali metals ion and negative silicate molecular ion arrival times which may indicate a de-coupling of the positive and negative ion formation process.

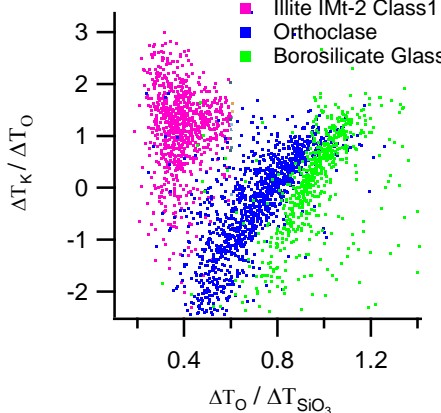

**Figure 11. Relative ion arrival times in positive ion with respected to relative ion arrival times in negative ion. Borosilicate glass and orthoclase feldspar polydisperse particles and the ISCM fraction (class 1) of illite IMt-2.**

### 4 Discussion

In TOF-MS the principal limitations in resolving power of an instrument are attributed to the differences in initial ion velocity distribution (energy focussing) and differences in the initial starting positions (space focussing); it is not easy to decouple these effects (Guilhaus, 1995). In addition, ion formation time, ion trajectory through the ion optics, and temporal jitter of the timing electronics all contribute to differences in arrival times of a certain ion species at the TOF-MS detector. In Matrix Assisted Laser Desorption Ionisation (MALDI), the sample is introduced on a sample plate in a fixed position. This means that initial ion velocity distributions are considered to be the primary cause of mass spectral peak broadening (Colby et al., 1994). In the case of SPMS, where the initial starting position is not fixed (due to particle beam divergence), space focussing is considered to be equally important as energy focussing in causing differences in ion arrival times.

Field free extraction may be a significant feature of the LAAP-TOF because space focussing is reduced to a simple difference in the time it takes an ion to enter the extraction optics, which is likely to be small compared to the effects of different ion velocities. In addition, the ion plume is allowed to evolve without the near instantaneous extraction of positive and negative ions that is a feature of systems using extraction by an electric field. One would expect a reduced plume density and therefore a reduction in space-charge effects and collisions in the ion plume as it is free to expand in all directions.





The co-axial geometry of the excimer laser with the particle beam is likely to introduce shot-to-shot variation in the position in which LDI takes place within the ion source and the commencement of ion formation with respect to the firing of the laser pulse. The location on the instrument axis at which a particle encounters the threshold fluence for LDI will vary with particle trajectory and the absorbing properties of the material.

Ion species dependence of the shift in ion arrival times recorded in the mass spectra for minerals dusts by the LAAP-TOF indicate that the shot to shot differences in average flight times of the ions is not a result of a temporal offset of the firing of the excimer laser and/or starting of the A/D timing device as this would affect all ion species equally. This reasoning leaves changes in initial ion velocity and ion formation time as the primary candidates for the cause of the peaks shifting and peak broadening observed but it is not possible to empirically derive initial ion velocity or ion formation times from the ion arrival

times alone. However, the relative differences in ion arrival times may hold clues to the nature of the ion formation mechanism even if the actual ion velocities and ion formation times are not quantified.

Measurement of initial kinetic energy of ions with MALDI indicated that the initial velocities of the matrix and analyte ions are identical, suggesting that the analyte molecule is entrained in to an expanding molecular jet of matrix ions and neutrals (Beavis and Chait, 1991; Pan and Cotter, 1992). The equal shift in ion arrival times of elemental and molecular ions

observed with borosilicate glass suggests that an equal addition to the scalar ion velocity and/or ion formation time, which can only be explained by shot to shot differences in ion formation time and initial ion velocities in a molecular jet. In contrast, the mass dependence to the negative ion peak shift for CB suggests a mass dependent velocity difference that could result from a thermal ionisation mechanism or a charge transfer and cluster decay ionisation mechanism, as suggested by Spengler & Kirsch, (2003) as a cause of mass dependent initial ion velocity differences in MALDI TOF.

It i that decay of the crystal lattice would be a factor in the ablation of mineral particles whose crystalline mineral structures have typical lattice energy of > 5000 kJmol$^{-1}$ (Jenkins et al., 2002), which far exceeds the energy available to a typical particle in a single laser pulse. Crystalline mineral structures could impose ion species dependence to the lattice decay and ion entrainment, such as that observed when comparing the average peak positions of the mineral dust with respect to the amorphous glass calibration. In clay minerals, the exchangeable interstitial cations that are weakly bonded

layer provide an energy sink for the laser energy and could be desorbed before the negatively changed tetrahedral and octahedral layers which then disintegrate by lattice decay. In this scenario, the effective de-coupling of the positive and negative ion formation, as suggested in the comparison of positive and negative ion arrival times (**Figure 11**), may result from differences in ion formation time of the $K^+$ and $SiO_3^-$ ions species. This process is not possible in feldspar mineral whose silicate structure must be broken in order to release the interstitial cation so that the $K^+$ and $SiO_3^-$ ion species coexist in the

ion plume, producing equal ion velocities due to coulombic forces and collisions.

The weak interaction of the interstitial complex with the silicate tetrahedra controls the stability of minerals in natural rock forming processes (Hawthorne, 2015) and would appear to have an influence on relative ion arrival times in SPMS. The influence of the interstitial potassium and sodium ion content on the relative arrival times of the $O^-$ and $SiO_3^-$ species forms the basis of our classification of mineral phase. Measurement of the potassium and sodium content by peak area



analysis is a potential source of uncertainty in the measurement due to particle matrix effects and the insufficient dynamic range of the TOF-MS detector. In addition, variation in the amount of energy encountered by particles due to instrument function and laser power setting could be an important consideration for the accuracy and reproducibility of the peak analysis. For example, one could postulate that the same initial ion velocities would be reached by all ions if enough pulse

energy is available to overcome the constraints of the lattice energy regardless of the crystal structure. However, it is precisely this shot to shot variation in energy and how the substance responds that gives rise to the distinct distribution in ion arrival times observed.

Provenance of the $O^-$ elemental ion in the negative ion spectra is a source of uncertainty in the interpretation of a lattice decay mechanism. In pure feldspars, the $O^-$ ion must be derived from the silica tetrahedra, but in clay minerals interstitial OH

molecules or absorbed water in the particles are additional sources of oxygen. The presence of water may be of significance as it is known to affect the ionisation process in LDI (Neubauer et al., 1998) and warrants further investigation. Regardless of the actual ionisation mechanism, careful calibration with a sample of known mineralogy, such as dry illite NX, will provide a suitable reference. The position of the convergence point, with respect to which the shift is measured, should be checked with illite NX after alignment of the excimer laser in the ion source.

**5 Conclusions**

A novel technique that uses peak centroid measurement in addition to peak areas to describe the mass spectral characteristics arising from the LDI of single particles of silicate mineral dust has been presented. To our knowledge, this is the first time that the properties of a material have been described by the relative changes in the ion arrival times of an ion species at a TOF-MS detector. Examination of the spectral patterns from dust samples reveals spectrum to spectrum

variation in the relative peak position of the $SiO_3^-$ molecular ion with respect to the $O^-$ elemental ion that occurs in distinct modes. Comparison of these modes with the borosilicate glass and carbon black suggest that the mode preference is a result of particle crystal structure and elemental composition, the properties that define mineral phase.

Analysis of clay mineral standards and nominally pure feldspars suggest that the relative shift of the elemental and molecular ions is a function of the quantity and co-ordination of potassium and sodium cations in the interstitial complex. It

is proposed that the mineral phase of the particle matrix influences the ion formation mechanism and produces variations in initial ion velocity and ion formation timing during the LDI of single particles. These effects are enhanced by the co-axial geometry of the excimer laser with the particle beam and are preserved in the field free extraction regime in the TOF-MS implemented in the LAAP-TOF. This may represent an important step in the understanding of how LDI proceeds in SPMS.

Analysis of multi-mineralic clay mineral standards reveals a multi-modal pattern in ion arrival times. A scheme that

classifies single particles has been defined on the basis of the alkali metal peak areas and the relative difference in the shift in the ion arrival times of the $O^-$ and $SiO_3^-$ species with respect to a calibration, a parameter we call $\tau$. Application of the



scheme to clay mineral standards result in the single particle differentiation of illite-smectite clays mineral (ISCM), feldspars and kaolinite that is in agreement with bulk mineralogy reported in semi-quantitative XRD analysis.

The nature of the interstitial complex and its effect on crystal structure can be extremely varied even within a single grain or crystal so that complete reproducibility would not be expected from any single particle measurement of a natural mineral dust sample. Most naturally occurring mineral dust particles are not homogenous in composition and it is expected that the result of the peak analysis to represent the principal matrix in clay sized particle (< 2 μm). In circumstances in which the actual mineral phase cannot be determined, it is still expected that the ions' arrival time ratios will be a useful parameter in describing differences in physiochemical properties of silicate particles in time resolved measurements. This represents an important step forward in the study of atmospheric processes where single particle mineral phase is important.

*Acknowledgements.* This work was supported by a PhD studentship awarded to N.Marsden by the Natural Environment Research Council (NERC). We would like to thank Alison Pawley, Kate Brodie and Merren Jones for their support with selecting representative mineral samples.

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
