# Peer review of "ON-LINE DIFFERENTIATION OF MINERAL PHASE IN AEROSOL PARTICLES BY ION FORMATION MECHANISM USING A LAAP-TOF SINGLE PARTICLE MASS SPECTROMETER"

_Atmospheric Measurement Techniques, 2017_

## Referee Comment (RC1) · Anonymous Referee #3 · 8 Aug 2017

General comments

The authors present a technique to analyze chemical composition and structure of airborne mineral dust particles using a laser ablation aerosol particle time-of-flight mass spectrometer (LAAPTOF), based on measured differences in key marker ion arrival times of chemically similar particle types, but with a variety of crystal structures. This is an interesting idea that merits publication. However, some sections could be written more clearly and/or need more information, the introduction could be somewhat streamlined, and some of the figures need higher resolution.

More specifically, I wonder about the usefulness of the analysis for other LAAPTOF

users given the difficulties in forming reproducible mass spectra with this instrument from complex atmospheric particles. The results presented here are based on one particular laser setting – are the effects e.g. supposed to scale linearly with laser power? I am aware that the authors might not be able to redo the experiments, but a discussion of the validity and transferability of results to other LAAPTOF (settings) should be added to the manuscript.

Another aspect that should be discussed, even if only in a speculative manner, is the applicability of results to ambient particle types that just contain fractions of dust, or are of more complex mixing state than the laboratory standards. The differentiation of clay mineral standards certainly indicates at least the potential for such studies to be performed with ambient samples, but this should be elaborated upon further.

Specific comments

P. 13, l. 5 – p. 14, l. 18: How was the mass calibration done, specifically? Was it performed on each raw spectrum individually, with resulting time series of parameters, which were subsequently averaged? Or were spectra averaged first, and then the calibration was performed? Also, have the authors explored a mass calibration with a 3-parameter fit, i.e a power law fit where the exponent is not kept at 0.5, and where the parameters are allowed to vary with time/spectra? This might actually reduce the shift in peak position in Figure 5. Both the mass scale and peak width dependence of the shift indicate failure of mass calibration. In other words, could one say that your study is in principle based on a failure of reproducible mass calibration in the LAAPTOF, and that you are using patterns of the failed mass calibration to infer mineral structure? What do your results signify for the mass calibration procedure in LAAPTOF in general?

Figure 6: Data points are based on how many spectra?

P. 18, l. 5/Table 7: Table 7 indicates a negative correlation between the interlayer charge and tau – can you show it graphically? The table is presented in the manuscript without much of an interpretation of the result. More negative interlayer charges seem

to increase "distortion" of spectra by reducing tau. Please elaborate further.

P. 19, l. 19 – 20: This paragraph would be easier to follow if you specified already here that you tried to classify the mineral samples, and that your results are number of particle per class.

P. 21, l. 6-12: The explanation of this part is too brief and should be expanded.

P. 22, l. 20: Can the authors say something about the influence of the size of ionization region on their results? If the plume expands in all directions, ions moving away from their respective extraction region of the bipolar TOF would presumably have a different flight time than ions moving towards their respective extraction region of the bipolar TOF, regardless of initial ion velocity.

P. 23, l. 1 – 19: It is not entirely clear in this paragraph if by "ion formation time" the authors here mean LDI, the time of particle-laser interaction, or specifically formation of individual ions within one specific particle. If they mean LDI, the shot-to-shot variation of LDI position based on particle flight time would influence ion formation time, and thus ion arrival time (which might be influenced non-linearly, depending on where ionization takes place, see comment above).

P. 24, l. 5-7: Is the shot-to-shot variation in energy delivered that large? The laser is presumably quite stable, and variation likely is more a question of how much energy is actually transferred to the particle, depending on when and where it is hit. This should be clarified.

Technical corrections

P. 2, l. 24: Closing bracket missing

P. 3, l. 10, and elsewhere: Physicochemical

P. 3, l.30: Typo, "because"

P. 4, l. 1: TOFMS is abbreviated TOF-MS in abstract

P. 4, l. 19 and throughout manuscript: LAAPTOF no dash

P. 12, l. 13: Typo, 2x "in"

Figure 6 doesn't print well

P. 17, l.19: Reference error

P. 19, l.19: Reference error, and weird sentence structure

P. 23, l. 15: Sentence structure ("that" is too much)

P. 23, l.20: Typos/sentence structure

P. 25, l. 5-7: Sentence structure

Figure 10: Specify what grey and red are.
* * *

---

## Referee Comment (RC2) · Anonymous Referee #2 · 12 Sep 2017

**Review of "ON-LINE DIFFERENTIATION OF MINERAL PHASE IN AEROSOL PARTICLES BY ION FORMATION MECHANISM USING A LAAPTOF SINGLE PARTICLE MASS SPECTROMETER" by Marsden et al.**

The authors present a novel way to qulitate minerals in single aerosol particles by type based on what seems to be reproducible matrix effects particular to the different mineral types. Although the manuscript could use a good proofread (see technical corrections below for some examples) it could also be expanded to note the reproducibility of these measurements. For instance it is unclear how sensitive the matrix effect is to various instrument parameters. Would the effects be particular to just the instrument in question or is it reproducible between instruments by the same manufacturer or between various aerosol mass spectrometric instruments. Without this information the applicability of this technique to the broader aerosol community is limited. However, if the method is indeed robust then this manuscript provides a step toward speciating aerosol particles by their mineral type. Finally the authors must address the real world applicability of this technique by including data on ambient aerosol if possible from a well-defined source. Real world data tests the limits of any instrumental procedure and can reveal how changing temperature, humidity, organic aerosol coatings, and heterogeneity in aerosol type, could affect the qualitative analysis presented in this paper. Any data that speaks to dependence of results on environmental parameters should be mentioned. At the end of the day this is a good manuscript worthy of publication so that others can help determine the extent to which this technique might be practical in a real world setting.

Other Major Corrections:
Pg 8 Ln 8-14. Can you speak to humidity effects on your measurements? does varying absolute humidity in the dust tower yield different results or matrix effects? Also the source of compressed air (company, and purity grade, water content) should be mentioned in the text
Pg 10 A couple of IGOR files/macros are mentioned however these seem to be homebuild analysis routines, The reader has no basis to judge the validity of these routines and thus they should be explained as to their function a bit more extensively, and/or code should be included in the supplemental if this hasn't already been done.
Pg 10 Define what is mean by "number of smoothes" and how the smoothing function works.
Pg 12 Ln 3 Why are the resolution of the TOF around the same resolution as a quadrupole mass filter. I would expect resolution of TOF to be in the 4000-5000 range. Please comment on the lack of resolution for your instrument

Technical Corrections:
Pg 2 ln 17 rephrase "The role of a mineral dust particle in the atmospheric processes…" to "The role of mineral dust particles in atmospheric processes"
Ln 20: remove "recently" as the articles cited are over 10 years old
Ln 24: need a closing parenthesis in the year 2010
Pg 3 Ln 9 define "NX" in "NX powder"
Pg 4 Ln 1 Be consistent with TOF-MS or TOFMS throughout the paper
Ln 18 change "markers ions" to "marker ions"
Pg 5 Ln 2 Be consistent with illite–smectite vs illite/smectite throughout the manuscript
Pg 10 Ln 11 please rephrase the sentence starting in "The ion…"
Figure 3 – please use a higher resolution image.
Figure 6 – reformat axis labels to be more readable
Pg 17 Ln 19 – remove "Error!..."
Pg 18 Ln 9 – remove extra space before the period.

Figure 9 – revise to clearly see the difference between the symbol types in part a.  In part b there is a typo on the y-axis label.  General quality of this figure needs to be improved upon.
Pg 23 Ln 20 revise for readability

---

## Author Response (AR1)

Dear Anonymous Referee #3,

We sincerely appreciate the detail with which you have reviewed our manuscript and the constructive comments given. Our response to your comments is given below.
On behalf of the authors,
Nick Marsden

*The response to the review is structured as follows: The original reviewer comments are given in black, followed by the author response in blue font colour.*

*Major corrections to the manuscript are reproduced in detail at the end of the this document.*

**Anonymous Referee #3**
()

General comments

The authors present a technique to analyze chemical composition and structure of airborne mineral dust particles using a laser ablation aerosol particle time-of-flight mass spectrometer (LAAPTOF), based on measured differences in key marker ion arrival times of chemically similar particle types, but with a variety of crystal structures. This is an interesting idea that merits publication. However, some sections could be written more clearly and/or need more information, the introduction could be somewhat streamlined, and some of the figures need higher resolution.

Introduction has been streamlined. See tracked-change document for details.

More specifically, I wonder about the usefulness of the analysis for other LAAPTOF users given the difficulties in forming reproducible mass spectra with this instrument from complex atmospheric particles. The results presented here are based on one particular laser setting – are the effects e.g. supposed to scale linearly with laser power? I am aware that the authors might not be able to redo the experiments, but a discussion of the validity and transferability of results to other LAAPTOF (settings) should be added to the manuscript.

Data was acquired at different excimer laser settings. This has been added to the manuscript as Section '3.3 The effect of laser power setting' (See Major correction A). This data suggests that differences in the distribution with respect to laser power setting may arise due to differences in ionisation threshold of different materials.

The Discussion section has been reorganised and now contains a discussion on the validity and transferability of the results to other LAAP-TOF (See Major Correction B). The tuning of the TOF ion optic is likely to have a big impact on the transferability to other LAAP-Tof, therefore the tune settings used for these experiments have been added to the supplement.

Another aspect that should be discussed, even if only in a speculative manner, is the applicability of results to ambient particle types that just contain fractions of dust, or are of more complex mixing state than the laboratory standards. The differentiation of clay mineral standards certainly indicates at least the potential for such studies to be performed with ambient samples, but this should be elaborated upon further.

The Discussion has been extended to a discussion of more complex particle such as desert dust and ambient sampling transported dust. The authors have analysed transported Saharan mineral dust which will be submitted for publication shortly, and is now referenced in the introduction and discussion sections (Marsden et al, manuscript in preparation, 2017). It is our intention that the current paper sets out the technical details of the method and the follow-up paper will demonstrate the application to ambient dust.

In addition, the ambient analysis is available in Chapter 8 of Nick Marsden's thesis, which will be available open access at the University of Manchester Library.

Specific comments

P. 13, l. 5 – p. 14, l. 18: How was the mass calibration done, specifically? Was it performed on each raw spectrum individually, with resulting time series of parameters, which were subsequently averaged? Or were spectra averaged first, and then the calibration was performed?

The procedure is explained in the methods section 2.3. .' Calibration coefficients were calculated for each sample type by fitting the first order approximation of the time of flight equation (Eq. 1) to the averaged peak centroid of three ion species'

I have clarified this in the table 6 caption   '*Table 1. Mass calibration coefficients calculated from the mode peak position (TOF) of 3 ion species for each samples in each ion mode.*'

Also, have the authors explored a mass calibration with a 3-parameter fit, i.e a power law fit where the exponent is not kept at 0.5, and where the parameters are allowed to vary with time/spectra? This might actually reduce the shift in peak position in Figure 5.

The OEM Data analysis software gives the option to make a 3-parameter fit on each spectrum individually, which reduces the peak shift as long as the peak position in ±0.5Da of the original calibration. Stick spectra are then created from the 3-paramter fit calibration.

Both the mass scale and peak width dependence of the shift indicate failure of mass calibration. In other words, could one say that your study is in principle based on a failure of reproducible mass calibration in the LAAPTOF, and that you are using patterns of the failed mass calibration to infer mineral structure? What do your results signify for the mass calibration procedure in LAAPTOF in general?

The mineral structure is inferred from the differences in the raw ion arrival times. Performing the mass calibration removes detail of the shot-to-shot variation. The first order calibration is performed to highlight the differences in behaviour of the samples and demonstrate the mass dependence on shifting. The impact of the mass scale calibration accuracy will be discussed in detail in the follow-up paper featuring ambient aerosol data.

Figure 6: Data points are based on how many spectra?

800 spectra per sample. This has been added to the caption.

P. 18, l. 5/Table 7: Table 7 indicates a negative correlation between the interlayer charge and tau – can you show it graphically? The table is presented in the manuscript without much of an interpretation of the result. More negative interlayer charges seem to increase "distortion" of spectra by reducing tau. Please elaborate further.

This is discussed in the first paragraph in section 3.4.
'There is some negative correlation between τ and the layer charge, with the exception of the calcium rich montmorillonite STx-1.'
Because calcium montmorillonite is the exception, I have plotted Na + K fraction from table 7 against tau in figure 9 (now figure 10) instead of the layer charge. I have modified the caption in figure 10 to make this clearer

P. 19, l. 19 – 20: This paragraph would be easier to follow if you specified already here that you tried to classify the mineral samples, and that your results are number of particle per class.

This paragraph has been altered to make it clearer that we are classifying particle numbers.

P. 21, l. 6-12: The explanation of this part is too brief and should be expanded.

This has been expanded to give a more detailed commentary of the results and more detailed discussion of the factors affecting the accuracy of the measurement.

P. 22, l. 20: Can the authors say something about the influence of the size of ionization region on their results? If the plume expands in all directions, ions moving away from their respective extraction region of the bipolar TOF would presumably have a different flight time than ions moving towards their respective extraction region of the bipolar TOF, regardless of initial ion velocity.

The first TOF extraction lens is grounded, and hence field free extraction. In such a there is no turn-around time associated with the TOF as only ions that have initial trajectories towards the respective positive and negative ion lenses are accepted. (as pointed out in the method section 2.3). The acceptance angle is not known to the authors but we expect the differences in TOF due to this is relatively small compared to the initial ion velocity.

P. 23, l. 1 – 19: It is not entirely clear in this paragraph if by "ion formation time" the authors here mean LDI, the time of particle-laser interaction, or specifically formation of individual ions within one specific particle. If they mean LDI, the shot-to-shot variation of LDI position based on particle flight time would influence ion formation time, and thus ion arrival time (which might be influenced non-linearly, depending on where ionization takes place, see comment above).

The ion formation time is a result of both processes suggested above. The following sentence has been added to this paragraph:
'Changes in ion formation time will include differences in the timing of the initial particle-laser interaction, due to particle trajectory and the properties of the material, as well as the timing of ion species formation after the ablation process has commenced.'
In the following paragraph, the possibility of the de-coupling of the formation of K+ and SiO3- in illite rich samples is suggested. This would be an example of both processes influencing the ion formation time. The possibility this occurs due to the co-axial geometry of the excimer laser is now mentioned in the final paragraph of the discussion.

P. 24, l. 5-7: Is the shot-to-shot variation in energy delivered that large? The laser is presumably quite stable, and variation likely is more a question of how much energy is

actually transferred to the particle, depending on when and where it is hit. This should be clarified.

This line has been removed during the reworking of the discussion section.

Technical corrections

P. 2, l. 24: Closing bracket missing
corrected
P. 3, l. 10, and elsewhere: Physicochemical
corrected
P. 3, l.30: Typo, "because"
corrected
P. 4, l. 1: TOFMS is abbreviated TOF-MS in abstract

corrected

P. 4, l. 19 and throughout manuscript: LAAPTOF no dash
corrected
P. 12, l. 13: Typo, 2x "in"
corrected
Figure 6 doesnot print well
corrected
P. 17, l.19: Reference error
P. 19, l.19: Reference error, and weird sentence structure
P. 23, l. 15: Sentence structure ("that" is too much)
corrected
P. 23, l.20: Typos/sentence structure
corrected
P. 25, l. 5-7: Sentence structure
corrected
Figure 10: Specify what grey and red are.

Now specified in the caption

**Major Correction A**

**3.3 The effect of laser power setting**

The laser power setting is important parameter in SPMS because, along with the size of the focal point and pulse duration, the amount of energy contained in each pulse defines the peak power density that occurs in the ionisation region. Differences in power density have been shown to affect the mass spectral patterns produced. For example, Reents and Schabel (2001) found that variation in peak power density, achieved by varying the 193 nm laser power setting, resulted in variations in the sodium fraction reported in the mass spectra of NaCl. The effect of different laser power setting on the distribution of $\tau$ values for kaolinite sample KGa-1b and illite NX is shown in **Figure 1**. For the kaolinite rich sample, increasing the pulse energy results in a narrowing of the distribution. The effect on the illite rich sample is somewhat different, in that increasing the pulse energy has the effect of increasing the number of particles in the mode $\tau > 1$.

[Figure]

**Figure 1.** *Histograms of the ion arrival times shift ratio ($\tau$) of the elemental ion O- and the molecular ion SiO_3-. with different laser pulse energy settings.(a) kaolinite sample KGa-1b and (b) illite NX.*

An explanation for this behaviour may be found in the hit-rate efficiency, defined as the number of spectra acquired divided by the number of particles detected (**Table 2**). The hit rate for illite NX was more than double that of KGa-1b at 4.5mJ. In addition, hit rate was much more sensitive to energy setting with KGa-1b than with illite NX. Increasing the laser energy from 2.8mJ to 7.6mJ per pulse resulted in a hit rate increase of 42% for the KGa-1b sample. Species dependent hit rates are associated with the power density threshold required to initiate the ablation/ionisation process, which is related to the lattice energy and absorbing properties of the material at 193nm (Thomson et al., 1997). It is therefore reasonable to assume that the increase in the $\tau > 1$ fraction in illite NX at 7.6mJ per pulse is partly due to the presence of a material with a high power density threshold such as kaolinite. The increase in hit rate with KGa-1b at 7.6mJ was in a large part due to the acquisition of spectra showing a purer form of kaolinite as indicated by the titanium content (**Table 2**). It is not known if the titanium was present in these particles as a structural cation, therefore altering the crystal structure, or as free titanium impurity.

**Table 2.** *The effect of the excimer pulse energy setting on the particle hit rate with illite NX and kaolinite sample KGa-1b. The number of particles that are low (<10% peak area) and high (>10% peak area) in titanium are given for the KGa-1b sample. The titanium content in the mass spectra was calculated from the combined peak area of the Ti$^+$ (m/z 48) and TiO$^+$ (m/z 64) and the total positive ion signal.*

| Sample | Excimer | Particles Detected | Spectra Acquired | Hit Rate | Ti < 10% | Ti > 10% |
|---|---|---|---|---|---|---|
| Illite NX | 2.8mJ | 7200 | 1901 | 0.26 | n/a | n/a |
| | 4.5mJ | 7200 | 2106 | 0.29 | n/a | n/a |
| | 7.6mJ | 7200 | 1825 | 0.25 | n/a | n/a |
| KGa-b1 | 2.8mJ | 7200 | 794 | 0.11 | 30 | 764 |
| | 4.5mJ | 7200 | 994 | 0.14 | 86 | 908 |
| | 7.6mJ | 7200 | 1397 | 0.19 | 673 | 724 |

**Major Correction B**

**4 Discussion**

In TOF-MS the principal limitations in resolving power of an instrument are attributed to the differences in initial ion velocity distribution (energy focussing) and differences in the initial starting positions (space focussing); it is not easy to decouple these effects (Guilhaus, 1995). In addition, ion formation time, ion trajectory through the ion optics, and temporal jitter of the timing electronics all contribute to differences in arrival times of a certain ion species at the TOF-MS detector. It is not possible to empirically derive starting position, initial ion velocity or ion formation times from the ion arrival times alone. However, the relative differences in ion arrival times may hold clues to the nature of the ion formation mechanism even if the actual ion velocities and ion formation times are not quantified.

Ion arrival times have been studied with the Matrix Assisted Laser Desorption Ionisation (MALDI) technique, where the sample is presented on a sample plate. With this method of sample introduction initial ion velocity distributions are considered to be the primary cause of mass spectral peak broadening (Colby et al., 1994) because the sample position in fixed, reducing the effects of space focussing. A measurement of initial kinetic energy of ions with MALDI indicated that the initial velocities of the matrix and analyte ions are identical, suggesting that the analyte molecule is entrained in to an expanding molecular jet of matrix ions and neutrals (Beavis and Chait, 1991; Pan and Cotter, 1992). In contrast, Spengler & Kirsch, (2003) observed a mass dependent initial ion velocity that could result from a thermal ionisation or a charge transfer and cluster decay ionisation mechanism.

In the case of SPMS, where the initial starting position is not fixed (due to particle beam divergence), space focussing is considered to be equally important as energy focussing in causing differences in ion arrival times. However, with the field free extraction featured in the LAAP-TOF, space focussing is reduced to a simple difference in the time it takes an ion to enter the extraction optics, which is likely to be small compared to the effects of different ion velocities. In addition, ion species dependence of the shift in ion arrival times recorded in the mass spectra indicate that the shot to shot differences in average flight times of the ions is not a result of a temporal offset of the firing of the excimer laser and/or starting of the A/D timing device as this would affect all ion species equally. This reasoning leaves changes in initial ion velocity and ion formation time as the primary candidates for the cause of the peaks shifting and peak broadening observed. Changes in ion formation time will include differences in the timing of the initial particle-laser interaction, due to particle trajectory and the properties of the material, as well as the timing of ion species formation after the ablation process has

commenced. The equal shift in ion arrival times of elemental and molecular ions observed with borosilicate glass suggests an equal addition to the scalar ion velocity and/or ion formation time, which can only be explained by shot to shot differences in ion formation time and initial ion velocities in a molecular jet. In contrast, the mass dependence to the negative ion peak shift for CB suggests a mass dependent velocity difference which suggests thermal ionisation or a charge transfer and cluster decay ionisation mechanism.

It is reasonable that decay of the crystal lattice would be a factor in the ablation of mineral particles whose crystalline mineral structures have typical lattice energy of $> 5000$ kJmol$^{-1}$ (Jenkins et al., 2002), which far exceeds the energy available to a typical particle in a single laser pulse. Crystalline mineral structures could impose ion species dependence to the lattice decay and ion entrainment, such as that observed when comparing the average peak positions of the mineral dust with respect to the amorphous glass calibration. In clay minerals, the exchangeable interstitial cations that are weakly bonded layer provide an energy sink for the laser energy and could be desorbed before the negatively changed tetrahedral and octahedral layers which then disintegrate by lattice decay. In this scenario, the effective de-coupling of the positive and negative ion formation, as suggested in the comparison of positive and negative ion arrival times (Figure 12), may result from differences in ion formation time and initial velocities of the $K^+$ and $SiO_3^-$ ions species. This process is not possible in feldspar mineral whose silicate structure must be broken in order to release the interstitial cation so that the $K^+$ and $SiO_3^-$ ion species coexist in the ion plume, producing equal ion velocities due to coulombic forces and collisions. The provenance of the $O^-$ elemental ion in the negative ion spectra is a source of uncertainty in the interpretation of a lattice decay mechanism. In pure feldspars, the $O^-$ ion must be derived from the silica tetrahedra, but in clay minerals interstitial OH molecules or absorbed water in the particles are additional sources of oxygen. The presence of water may be of significance as it is known to affect the ionisation process in LDI (Neubauer et al., 1998) and warrants further investigation.

The weak interaction of the interstitial complex with the silicate tetrahedra controls the stability of minerals in natural rock forming processes (Hawthorne, 2015) and would appear to have an influence on relative ion arrival times in SPMS. The influence of the interstitial potassium and sodium ion content on the relative arrival times of the $O^-$ and $SiO_3^-$ species forms the basis of our classification of mineral phase. Measurement of the potassium and sodium content by peak area analysis is a potential source of uncertainty in the measurement due to particle matrix effects and the insufficient dynamic range of the TOF-MS detector. In addition, the amount of energy encountered by particles due to instrument function and laser power setting could be an important consideration for the accuracy and reproducibility of the analysis. The influence of laser power setting on the hit rate for kaolinite and illite demonstrate the potential for number fraction bias in the classification. Using the highest leaser power setting would not be desirable for ambient sampling because of excessive fragmentation of non-silicate material such as internally and externally mixed organic material. One could also postulate that the same initial ion velocities would be reached by all ions if enough pulse energy is available to overcome the constraints of the lattice energy regardless of the crystal structure. In this study we found that 4.5mJ/pulse was a suitable laser setting for differentiating particles types in illite NX.

The differentiation of mineral phases in this study was demonstrated with clay mineral standards with well characterised composition. The classification system was defined using the mineralogical composition from XRD analysis as a guide. Because of the huge variety in mineral phase that occurs in nature and the potential for mixing of phase within a single particle, the identification of specific mineral phases is expected to be more

difficult in complex natural samples such as desert dust, especially if XRD analysis of mineralogy is not available for reference. Even so, the distribution of τ values is expected to provide insight into the composition of clay sized fraction of a dust sample even if the exact mineral phase is not clearly identified. Analysis of the mineralogy of transported Saharan dust measured at Praia, Cape Verde Islands during the ICE-D campaign (August 2015) is presented in a separate publication (Marsden *et al*, manuscript in preparation, 2017)

Further work is required in development of this method. The tuning of the ion optics is of particularly important in determining the flight times of ions with respect to initial ion velocity. Modelling of the ion trajectories in a software package such as SIMION (Scientific Instrument Services, Inc.) may provide optimised tuning that further exploits the differences in ion focussing that arises from differences in particle composition. The transferability of the method to other LAAP-ToF instruments will also likely depend on the tuning of the ion optics. Tuning parameters used for this study are provided in the supplement (S1). Establishing the role of fixed instrument design features will indicate how transferable this method is to other designs of single particle mass spectrometers. For example, using extraction by an electric field one would expect an increased plume density and therefore an increase in space-charge effects and collisions as the ion plume is not free to expand in all directions. In addition, an orthogonal geometry of the excimer laser with the particle beam is likely to produce less variation in the position and timing in which a particle encounters the threshold power density for LDI compared to the co-axial geometry used in the LAAP-ToF.

We sincerely appreciate the detail with which you have reviewed our manuscript and the constructive comments given. Our response to your comments is given below.
On behalf of the authors,
Nick Marsden

*The response to the review is structured as follows: The original reviewer comments are given in black, followed by the author response in blue font colour.*

*Major corrections to the manuscript are reproduced in detail at the end of the document.*

**Anonymous Referee #2**
()

The authors present a novel way to qulitate minerals in single aerosol particles by type based on what seems to be reproducible matrix effects particular to the different mineral types. Although the manuscript could use a good proofread (see technical corrections below for some examples) it could also be expanded to note the reproducibility of these measurements. For instance it is unclear how sensitive the matrix effect is to various instrument parameters. Would the effects be particular to just the instrument in question or is it reproducible between instruments by the same manufacturer or between various aerosol mass spectrometric instruments. Without this information the applicability of this technique to the broader aerosol community is limited. However, if the method is indeed robust then this manuscript provides a step toward speciating aerosol particles by their mineral type.

Data that shows the reproducibility at different laser power settings has been added to the manuscript in Section 3.3 'The effect of laser power setting' (See Major Correction A)

The Discussion section has been reorganised and now contains a discussion on the validity and transferability of the results to other LAAP-TOF (See Major Correction B). The tuning of the TOF ion optic is likely to have a big impact on the transferability to other LAAP-Tof, therefore the tune settings used for these experiments have been added to the supplement. The transferability to other instrument models is now discussed towards the end of the Discussion section. The role of the co-axial geometry of the excimer laser and the field free extraction may be important design features.

Finally the authors must address the real world applicability of this technique by including data on ambient aerosol if possible from a well-defined source. Real world data tests the limits of any instrumental procedure and can reveal how changing temperature, humidity, organic aerosol coatings, and heterogeneity in aerosol type, could affect the qualitative analysis presented in this paper. Any data that speaks to dependence of results on environmental parameters should be mentioned. At the end of the day this is a good manuscript worthy of publication so that others can help determine the extent to which this technique might be practical in a real world setting.

The Discussion section has been extended to include more complex particle such as desert dust and ambient sampling transported dust. The authors have analysed transported Saharan mineral dust which will be submitted for publication shortly, and is now referenced in the introduction and discussion sections (Marsden et al, manuscript in preparation, 2017). It is our intention that the current paper sets out the technical details of the method and the follow-up paper will demonstrate the application to ambient dust with reference to humidity and mixing state.

In addition, the ambient analysis is available in Chapter 8 of Nick Marsden thesis, which will be available open access at the University of Manchester Library.

Other Major Corrections:
Pg 8 Ln 8-14. Can you speak to humidity effects on your measurements? does varying absolute humidity in the dust tower yield different results or matrix effects? Also the source of compressed air (company, and purity grade, water content) should be mentioned in the text

The humidity was not varied during the experiments.
The following line has been added to Section 2.2 Experimental Setup 'adjustable flow of dried, filtered and oil free compressed air (produced onsite using a compressor) introduced into the bottom of the tower'.

Pg 10 A couple of IGOR files/macros are mentioned however these seem to be homebuild analysis routines, The reader has no basis to judge the validity of these routines and thus they should be explained as to their function a bit more extensively, and/or code should be included in the supplemental if this hasn't already been done.
The peak fitting was carried out using a procedure (multipeak fit v2) in a commercially available software package igor v6.36. The only homebuilt part of this was used to apply this procedure to my data in batches. In our view it is not necessary to provide the code for this simple procedure in the supplement.

Pg 10 Define what is mean by "number of smoothes" and how the smoothing function works.
The number of smoothes is simply the data smoothing parameterisation in the commercially available software noted above.
Pg 12 Ln 3 Why are the resolution of the TOF around the same resolution as a quadrupole mass filter. I would expect resolution of TOF to be in the 4000-5000 range. Please comment on the lack of resolution for your instrument
The resolution produced by our instrument is typical in SPMS due to space focussing and energy focussing difficulties associated with this technique. See (Murphy, 2007) and references therein for further details.

Technical Corrections:
Pg 2 ln 17 rephrase "The role of a mineral dust particle in the atmospheric processes…" to "The role of mineral dust particles in atmospheric processes"
corrected
Ln 20: remove "recently" as the articles cited are over 10 years old
corrected
Ln 24: need a closing parenthesis in the year 2010
corrected
Pg 3 Ln 9 define "NX" in "NX powder"
This is brand name for a product. The definition is unknown to the authors.
Pg 4 Ln 1 Be consistent with TOF-MS or TOFMS throughout the paper

corrected

Ln 18 change "markers ions" to "marker ions"

corrected

Pg 5 Ln 2 Be consistent with illite–smectite vs illite/smectite throughout the manuscript

Corrected to illite-smectite

Pg 10 Ln 11 please rephrase the sentence starting in "The ion…"

Corrected to 'The configuration of the extraction optics in the LAAP-TOF is unusual in that the first extraction plate is grounded resulting in field free ionisation region.'

Figure 3 – please use a higher resolution image.

corrected

Figure 6 – reformat axis labels to be more readable

corrected

Pg 17 Ln 19 – remove "Error!..."

corrected

Pg 18 Ln 9 – remove extra space before the period.

Figure 9 – revise to clearly see the difference between the symbol types in part a. In part b there is a typo on the y-axis label. General quality of this figure needs to be improved upon.

Pg 23 Ln 20 revise for readability
corrected

**References**

Murphy, D. M.: The design of single particle laser mass spectrometers, Mass Spectrom. Rev., 26, 150–165, doi:10.1002/mas, 2007.

**Major Correction A**

**3.3 The effect of laser power setting**

The laser power setting is important parameter in SPMS because, along with the size of the focal point and pulse duration, the amount of energy contained in each pulse defines the peak power density that occurs in the ionisation region. Differences in power density have been shown to affect the mass spectral patterns produced. For example, Reents and Schabel (2001) found that variation in peak power density, achieved by varying the 193 nm laser power setting, resulted in variations in the sodium fraction reported in the mass spectra of NaCl. The effect of different laser power setting on the distribution of $\tau$ values for kaolinite sample KGa-1b and illite NX is shown in **Figure 1**. For the kaolinite rich sample, increasing the pulse energy results in a narrowing of the distribution. The effect on the illite rich sample is somewhat different, in that increasing the pulse energy has the effect of increasing the number of particles in the mode $\tau > 1$.

[Figure]

**Figure 1.** *Histograms of the ion arrival times shift ratio ($\tau$) of the elemental ion O- and the molecular ion $SiO_3$- with different laser pulse energy settings.(a) kaolinite sample KGa-1b and (b) illite NX.*

An explanation for this behaviour may be found in the hit-rate efficiency, defined as the number of spectra acquired divided by the number of particles detected (**Table 1**). The hit rate for illite NX was more than double that of KGa-1b at 4.5mJ. In addition, hit rate was much more sensitive to energy setting with KGa-1b than with illite NX. Increasing the laser energy from 2.8mJ to 7.6mJ per pulse resulted in a hit rate increase of 42% for the KGa-1b sample. Species dependent hit rates are associated with the power density threshold required to initiate the ablation/ionisation process, which is related to the lattice energy and absorbing properties of the material at 193nm (Thomson et al., 1997). It is therefore reasonable to assume that the increase in the $\tau > 1$ fraction in illite NX at 7.6mJ per pulse is partly due to the presence of a material with a high power density threshold such as kaolinite. The increase in hit rate with KGa-1b at 7.6mJ was in a large part due to the acquisition of spectra showing a purer form of kaolinite as indicated by the titanium content (**Table 1**). It is not known if the titanium was present in these particles as a structural cation, therefore altering the crystal structure, or as free titanium impurity.

**Table 1.** *The effect of the excimer pulse energy setting on the particle hit rate with illite NX and kaolinite sample KGa-1b. The number of particles that are low (<10% peak area) and high (>10% peak area) in titanium are given for the KGa-1b sample. The titanium content in the mass spectra was calculated from the combined peak area of the $Ti^+$ (m/z 48) and $TiO^+$ (m/z 64) and the total positive ion signal.*

| Sample | Excimer | Particles Detected | Spectra Acquired | Hit Rate | Ti < 10% | Ti > 10% |
|--------|---------|-------------------|------------------|----------|----------|----------|
| Illite NX | 2.8mJ | 7200 | 1901 | 0.26 | n/a | n/a |
| | 4.5mJ | 7200 | 2106 | 0.29 | n/a | n/a |
| | 7.6mJ | 7200 | 1825 | 0.25 | n/a | n/a |
| KGa-b1 | 2.8mJ | 7200 | 794 | 0.11 | 30 | 764 |
| | 4.5mJ | 7200 | 994 | 0.14 | 86 | 908 |
| | 7.6mJ | 7200 | 1397 | 0.19 | 673 | 724 |

**Major Correction B**

**4 Discussion**

In TOF-MS the principal limitations in resolving power of an instrument are attributed to the differences in initial ion velocity distribution (energy focussing) and differences in the initial starting positions (space focussing); it is not easy to decouple these effects (Guilhaus, 1995). In addition, ion formation time, ion trajectory through the ion optics, and temporal jitter of the timing electronics all contribute to differences in arrival times of a certain ion species at the TOF-MS detector. It is not possible to empirically derive starting position, initial ion velocity or ion formation times from the ion arrival times alone. However, the relative differences in ion arrival times may hold clues to the nature of the ion formation mechanism even if the actual ion velocities and ion formation times are not quantified.

Ion arrival times have been studied with the Matrix Assisted Laser Desorption Ionisation (MALDI) technique, where the sample is presented on a sample plate. With this method of sample introduction initial ion velocity distributions are considered to be the primary cause of mass spectral peak broadening (Colby et al., 1994) because the sample position in fixed, reducing the effects of space focussing. A measurement of initial kinetic energy of ions with MALDI indicated that the initial velocities of the matrix and analyte ions are identical, suggesting that the analyte molecule is entrained in to an expanding molecular jet of matrix ions and neutrals (Beavis and Chait, 1991; Pan and Cotter, 1992). In contrast, Spengler & Kirsch, (2003) observed a mass dependent initial ion velocity that could result from a thermal ionisation or a charge transfer and cluster decay ionisation mechanism.

In the case of SPMS, where the initial starting position is not fixed (due to particle beam divergence), space focussing is considered to be equally important as energy focussing in causing differences in ion arrival times. However, with the field free extraction featured in the LAAP-TOF, space focussing is reduced to a simple difference in the time it takes an ion to enter the extraction optics, which is likely to be small compared to the effects of different ion velocities. In addition, ion species dependence of the shift in ion arrival times recorded in the mass spectra indicate that the shot to shot differences in average flight times of the ions is not a result of a temporal offset of the firing of the excimer laser and/or starting of the A/D timing device as this would affect all ion species equally. This reasoning leaves changes in initial ion velocity and ion formation time as the primary candidates for the cause of the peaks shifting and peak broadening observed. Changes in ion formation time will include differences in the timing of the initial particle-laser interaction, due to particle trajectory and the

properties of the material, as well as the timing of ion species formation after the ablation process has commenced. The equal shift in ion arrival times of elemental and molecular ions observed with borosilicate glass suggests an equal addition to the scalar ion velocity and/or ion formation time, which can only be explained by shot to shot differences in ion formation time and initial ion velocities in a molecular jet. In contrast, the mass dependence to the negative ion peak shift for CB suggests a mass dependent velocity difference which suggests thermal ionisation or a charge transfer and cluster decay ionisation mechanism.

It is reasonable that decay of the crystal lattice would be a factor in the ablation of mineral particles whose crystalline mineral structures have typical lattice energy of $> 5000$ kJmol$^{-1}$ (Jenkins et al., 2002), which far exceeds the energy available to a typical particle in a single laser pulse. Crystalline mineral structures could impose ion species dependence to the lattice decay and ion entrainment, such as that observed when comparing the average peak positions of the mineral dust with respect to the amorphous glass calibration. In clay minerals, the exchangeable interstitial cations that are weakly bonded layer provide an energy sink for the laser energy and could be desorbed before the negatively changed tetrahedral and octahedral layers which then disintegrate by lattice decay. In this scenario, the effective de-coupling of the positive and negative ion formation, as suggested in the comparison of positive and negative ion arrival times (Figure 12), may result from differences in ion formation time and initial velocities of the $K^+$ and $SiO_3^-$ ions species. This process is not possible in feldspar mineral whose silicate structure must be broken in order to release the interstitial cation so that the $K^+$ and $SiO_3^-$ ion species coexist in the ion plume, producing equal ion velocities due to coulombic forces and collisions. The provenance of the $O^-$ elemental ion in the negative ion spectra is a source of uncertainty in the interpretation of a lattice decay mechanism. In pure feldspars, the $O^-$ ion must be derived from the silica tetrahedra, but in clay minerals interstitial OH molecules or absorbed water in the particles are additional sources of oxygen. The presence of water may be of significance as it is known to affect the ionisation process in LDI (Neubauer et al., 1998) and warrants further investigation.

The weak interaction of the interstitial complex with the silicate tetrahedra controls the stability of minerals in natural rock forming processes (Hawthorne, 2015) and would appear to have an influence on relative ion arrival times in SPMS. The influence of the interstitial potassium and sodium ion content on the relative arrival times of the $O^-$ and $SiO_3^-$ species forms the basis of our classification of mineral phase. Measurement of the potassium and sodium content by peak area analysis is a potential source of uncertainty in the measurement due to particle matrix effects and the insufficient dynamic range of the TOF-MS detector. In addition, the amount of energy encountered by particles due to instrument function and laser power setting could be an important consideration for the accuracy and reproducibility of the analysis. The influence of laser power setting on the hit rate for kaolinite and illite demonstrate the potential for number fraction bias in the classification. Using the highest leaser power setting would not be desirable for ambient sampling because of excessive fragmentation of non-silicate material such as internally and externally mixed organic material. One could also postulate that the same initial ion velocities would be reached by all ions if enough pulse energy is available to overcome the constraints of the lattice energy regardless of the crystal structure. In this study we found that 4.5mJ/pulse was a suitable laser setting for differentiating particles types in illite NX.

The differentiation of mineral phases in this study was demonstrated with clay mineral standards with well characterised composition. The classification system was defined using the mineralogical composition from XRD analysis as a guide. Because of the huge variety in mineral phase that occurs in nature and the potential for

mixing of phase within a single particle, the identification of specific mineral phases is expected to be more difficult in complex natural samples such as desert dust, especially if XRD analysis of mineralogy is not available for reference. Even so, the distribution of $\tau$ values is expected to provide insight into the composition of clay sized fraction of a dust sample even if the exact mineral phase is not clearly identified. Analysis of the mineralogy of transported Saharan dust measured at Praia, Cape Verde Islands during the ICE-D campaign (August 2015) is presented in a separate publication (Marsden *et al*, manuscript in preparation, 2017)

Further work is required in development of this method. The tuning of the ion optics is of particularly important in determining the flight times of ions with respect to initial ion velocity. Modelling of the ion trajectories in a software package such as SIMION (Scientific Instrument Services, Inc.) may provide optimised tuning that further exploits the differences in ion focussing that arises from differences in particle composition. The transferability of the method to other LAAP-ToF instruments will also likely depend on the tuning of the ion optics. Tuning parameters used for this study are provided in the supplement (S1). Establishing the role of fixed instrument design features will indicate how transferable this method is to other designs of single particle mass spectrometers. For example, using extraction by an electric field one would expect an increased plume density and therefore an increase in space-charge effects and collisions as the ion plume is not free to expand in all directions. In addition, an orthogonal geometry of the excimer laser with the particle beam is likely to produce less variation in the position and timing in which a particle encounters the threshold power density for LDI compared to the co-axial geometry used in the LAAP-ToF.

[revised manuscript text omitted]
 processing software provides an additional facility to apply a unique 3 parameter fit calibration  to each spectrum. However, for this calibration to be successful, the peak of interest must be located $\pm 0.5$ Da from the average mass position which is not the case in a significant number of peaks in mineral dust spectra.

The mass scale dependence of the peak position shift suggests that the differences in calibrations do not arise from a simple linear shift in ion arrival times for all ion species in the mass spectra. This is further demonstrated by examining $\Delta T_{(i)}$ with respect to the negative ion calibration for borosilicate glass (Fig. 6). Carbon black has a clear mass dependence with respect to this calibration. Kaolinite does not have a strong mass dependence but is affected by a large kink in the mass scale between the $SiO_2$ and $SiO_3$ molecular ions (m/z 60 and 76 respectively), which is apparent with all the silicate minerals analysed.

**3.2 Relative shift of elemental and molecular ion ToF.**

In this section we explore the relationship between the ion arrival times of $O^-$ elemental ion ($T_O$) and the $SiO_3^-$ molecular ion ($T_{SiO_3}$) within discrete ionisation events. In the case of borosilicate glass, an amorphous silicate material, a scatter plot of arrival times measured in individual mass spectra display an approximately linear distribution of data points with a gradient of 1 (Fig. 7a) indicating that $\Delta T_O \approx \Delta T_{SiO_3}$. In contrast, the non-silicate carbon black, a similar plot of elemental vs molecular carbon ions measured from C at m/z -12 and $C_6$ ion at m/z -60 respectively, gives a linear distribution of ion arrival times with a gradient of 0.35, indicating that generally $\Delta T_C \approx 0.35 * \Delta T_{C_6}$.

Similar distributions of ion arrival times can be observed with the LDI of crystalline silicate mineral dust. The framework silicate orthoclase feldspar, has a distribution of ion arrival times that is similar to the borosilicate glass except the distribution is narrower (Fig. 7b), whereas illite IMt-2, a sheet silicate, has a large mode that is similar to the carbon black, with an additional mode of particles that is more similar to the borosilicate glass. The silicate mineral sample with the most variation in ion arrival times is the illite NX where several distinct modes are apparent (Fig. 7c). The multimodal nature of this distribution was still apparent after the particles were mass selected with the CPMA before analysis with the LAAP-TOF. The ion arrival time distribution of mass selected 350 fg illite NX particle shows a smaller

variation of up to 20ns whereas mass selected 700 fg illite NX particles have a distribution comparable to the polydisperse analysis.

In order to directly compare the ion arrival time distributions of different silicate minerals, the shift in arrival time ($\Delta T_{(i)}$) was calculated for each spectrum with respect to a set point on the time scale. For silicate containing particles, $\Delta T_O$ and $\Delta T_{SiO_3}$ were calculated with respect to the point where the mode distributions converge on the scatter plots at $T_O = 4010.2$ ns and $T_{SiO_3} = 8722.4$ ns respectively, which we will call the convergence point. The relative difference in the ion arrival times of the elemental and molecular ions in negative ion mode can then be expressed as a ratio ($\tau$) for each particle analysed.

$$\tau = \frac{\Delta T_O}{\Delta T_{SiO_3}}$$

Histograms of $\tau$ values measured from the peak analysis in silicate mineral mass spectra are shown in Fig. 8. Potassium rich clay minerals illite NX and illite IMt-2 ((Fig. 8a) have a distinct mode around 0.3-0.5, that is aligned with the $\Delta T_C/\Delta T_{C_6}$ ratio derived from the peak analysis of carbon black. Other distinct modes can be seen in the illite IMt-2 sample at 0.79 and in the illite NX sample at 0.83 and 1.10 that is more similar to the distribution of $\tau$ values measured for borosilicate glass. The smectite and montmorillonite clays have modes in the distribution that range from 0.4 to 0.86 and the 1:1 layer clay kaolinite has the largest mode at 0.93 ((Fig. 8b).

**3.3 The effect of laser power setting**

The laser power setting is important parameter in SPMS because, along with the size of the focal point and pulse duration, the amount of energy contained in each pulse defines the peak power density that occurs in the ionisation region. Differences in power density have been shown to affect the mass spectral patterns produced. For example, Reents and Schabel (2001) found that variation in peak power density, achieved by varying the 193 nm laser power setting, resulted in variations in the sodium fraction reported in the mass spectra of NaCl. The effect of different laser power setting on the distribution of $\tau$ values for kaolinite.  sample KGa-1b and illite NX is shown in Fig 9. For the kaolinite rich sample, increasing the pulse energy results in a narrowing of the distribution. The effect on the illite rich sample is somewhat different, in that increasing the pulse energy has the effect of increasing the number of particles in the mode $\tau > 1$.

An explanation for this behaviour may be found in the hit-rate efficiency, defined as the number of spectra acquired divided by the number of particles detected (Table 7). The hit rate for illite NX was more than double that of KGa-1b at 4.5mJ. In addition, hit rate was much more sensitive to energy setting with KGa-1b than with illite NX. Increasing the laser energy from 2.8mJ to 7.6mJ per pulse resulted in a hit rate increase of 42% for the KGa-1b sample. Species dependent hit rates are associated with the power density threshold required to initiate the ablation/ionisation process, which is related to the lattice energy and absorbing properties of the material at 193nm (Thomson et al., 1997). It is therefore reasonable to assume that the increase in the $\tau > 1$ fraction in illite NX at 7.6mJ per pulse is partly due to the presence of a material with a high power density threshold such as kaolinite. The increase in hit rate with KGa-1b at 7.6mJ was in a large part due to the acquisition of spectra showing a purer form of kaolinite as indicated by the titanium content (Table 7). It is not known if the titanium was present in these particles as a structural cation, therefore altering the crystal structure, or as free titanium impurity.

Table 7. The effect of the excimer pulse energy setting on the particle hit rate with illite NX and kaolinite sample KGa-1b. The number of particles that are low (<10% peak area) and high (>10% peak area) in titanium are given for the KGa-1b sample. The titanium content in the mass spectra was calculated from the $\Delta T_C/\Delta T_{t_C}$ ratio.combined peak area of the Ti$^+$ (m/z 48) and TiO$^+$ (m/z 64) and the total positive ion signal.

| Sample | Excimer | Particles Detected | Spectra Acquired | Hit Rate | Ti < 10% | Ti > 10% |
|---|---|---|---|---|---|---|
| Illite NX | 2.8mJ | 7200 | 1901 | 0.26 | n/a | n/a |
| | 4.5mJ | 7200 | 2106 | 0.29 | n/a | n/a |
| | 7.6mJ | 7200 | 1825 | 0.25 | n/a | n/a |
| KGa-b1 | 2.8mJ | 7200 | 794 | 0.11 | 30 | 764 |
| | 4.5mJ | 7200 | 994 | 0.14 | 86 | 908 |
| | 7.6mJ | 7200 | 1397 | 0.19 | 673 | 724 |

**3.4 The role of the interstitial complex**

[revised manuscript text omitted]

The differentiation of mineral phase by this classification method has some broad agreement with the XRD analysis. The relative particle number fraction of ISCM between the three samples is in agreement with the relative mass concentration of illite derived from XRD i.e. ISCz-1 is the most illite rich followed by IMt-2 and illite NX. A greater fraction of feldspar and kaolinite is also measured for illite NX than for the other samples in agreement with XRD analysis. The main difference in the mineral fraction reported by XRD analysis and this peak analysis technique is the relative fraction of ISCM minerals with non-ISCM minerals such as K feldspar and kaolinite.mineral classes. The differences may arise from comparing a number counting technique with a mass fraction, and from hit-rate bias in the LAAP-TOF measurement which may discriminate against certain particle types due to morphology and composition.as discussed in section 3.3 above. In addition, the XRD analysis technique does not report the unidentified fraction of such as amorphous glassy material that may contribute to the non-ISCM fraction in the peak analysis and the degree of homogeneity with regards to mineral fractions in the bulk samples is not well characterised and may result in sample to sample and batch to batch differences in the composition of a clay mineral standard.

The relatively poor resolution of the mineral differentiation makes the identification of the exact mineral phase difficult. Whilst the plots of $\tau$ vs K + Na content (Fig. 11, top panels) produces distinct particle clusters for illite NX and IMt-2, the corresponding plot for ISCz-1 is much less differentiated. It is not clear if this occurs because of the poor resolution of the method or because the natural variation in composition of the latter sample.

**3.5 Relative shift of positive and negative ions**

The ratio $\Delta T_K/\Delta T_O$ is plotted against $\Delta T_O/\Delta T_{SiO_3}$ ($\tau$), in Figure 11.Fig. 12. In this plot, the distribution of data points for borosilicate glass and orthoclase feldspar suggest a positive correlation between these ratios, with the trends in the distributions converging towards a point where the magnitude of shift of all species is equal. In illite IMt-2 the ISCM mineral fraction, identified as class 1 in the classification scheme, does not show a correlation between positive alkali metals ion and negative silicate molecular ion arrival times which may indicate a de-coupling of the positive and negative ion formation process.

**4 Discussion**

In TOF-MS the principal limitations in resolving power of an instrument are attributed to the differences in initial ion velocity distribution (energy focussing) and differences in the initial starting positions (space focussing); it is not easy to decouple these effects (Guilhaus, 1995). In addition, ion formation time, ion trajectory through the ion optics, and temporal jitter of the timing electronics all contribute to differences in arrival times of a certain ion species at the TOF-MS detector.  It is not possible to empirically derive starting position,

~~Field free extraction may be a significant feature of the LAAP TOF because space focussing is reduced to a simple difference in the time it takes an ion to enter the extraction optics, which is likely to be small compared to the effects of different ion velocities. In addition, the ion plume is allowed to evolve without the near instantaneous extraction of positive and negative ions that is a feature of systems using extraction by an electric field. One would expect a reduced plume density and therefore a reduction in space charge effects and collisions in the ion plume as it is free to expand in all directions.~~

~~The co-axial geometry of the excimer laser with the particle beam is likely to introduce shot-to-shot variation in the position in which LDI takes place within the ion source and the commencement of ion formation with respect to the firing of the laser pulse. The location on the instrument axis at which a particle encounters the threshold fluence for LDI will vary with particle trajectory and the absorbing properties of the material.~~

~~Ion species dependence of the shift in ion arrival times recorded in the mass spectra for minerals dusts by the LAAP-TOF indicate that the shot to shot differences in average flight times of the ions is not a result of a temporal offset of the firing of the excimer laser and/or starting of the A/D timing device as this would affect all ion species equally. This reasoning leaves changes in initial ion velocity and ion formation time as the primary candidates for the cause of the peaks shifting and peak broadening observed but it is not possible to empirically derive~~ initial ion velocity or ion formation times from the ion arrival times alone. However, the relative differences in ion arrival times may hold clues to the nature of the ion formation mechanism even if the actual ion velocities and ion formation times are not quantified.

 Ion arrival times have been studied with the Matrix Assisted Laser Desorption Ionisation (MALDI) technique, where the sample is presented on a sample plate. With this method of sample introduction initial ion velocity distributions are considered to be the primary cause of mass spectral peak broadening (Colby et al., 1994) because the sample position in fixed, reducing the effects of space focussing. A measurement of initial kinetic energy of ions with MALDI indicated that the initial velocities of the matrix and analyte ions are identical,

suggesting that the analyte molecule is entrained in to an expanding molecular jet of matrix ions and neutrals (Beavis and Chait, 1991; Pan and Cotter, 1992). In contrast, Spengler & Kirsch, (2003) observed a mass dependent initial ion velocity that could result from a thermal ionisation or a charge transfer and cluster decay ionisation mechanism.

[revised manuscript text omitted]

The weak interaction of the interstitial complex with the silicate tetrahedra controls the stability of minerals in natural rock forming processes (Hawthorne, 2015) and would appear to have an influence on relative ion arrival times in SPMS. The influence of the interstitial potassium and sodium ion content on the relative arrival times of the O⁻ and SiO₃⁻ species forms the basis of our classification of mineral phase. Measurement of the potassium and sodium content by peak area analysis is a potential source of uncertainty in the measurement due to particle matrix effects and the insufficient dynamic range of the TOF-MS detector. In addition, the amount of energy encountered by particles due to instrument function and laser power setting could be an important consideration for the accuracy and reproducibility of the analysis. The influence of laser power setting on the hit rate for kaolinite and illite demonstrate the potential for number fraction bias in the classification. Using the highest leaser power setting would not be desirable for ambient sampling because of excessive fragmentation of non-silicate material such as internally and externally mixed organic material. One could also postulate that the same initial ion velocities would be reached by all ions if enough pulse energy is available to overcome the constraints of the lattice energy regardless of the crystal structure. In this study we found that 4.5mJ/pulse was a suitable laser setting for differentiating particles types in illite NX.

The differentiation of mineral phases in this study was demonstrated with clay mineral standards with well characterised composition. The classification system was defined using the mineralogical composition from XRD analysis as a guide. Because of the huge variety in mineral phase that occurs in nature and the potential for mixing of phase within a single particle, the identification of specific mineral phases is expected to be more difficult in complex natural samples such as desert dust, especially if XRD analysis of mineralogy is not available for reference. Even so, the distribution of $\tau$ values is expected to provide insight into the composition of clay sized fraction of a dust sample even if the exact mineral phase is not clearly identified. Analysis of the mineralogy of transported Saharan dust measured at Praia, Cape Verde Islands during the ICE-D campaign (August 2015) is presented in a separate publication (Marsden *et al*, manuscript in preparation, 2017)

Further work is required in development of this method. The tuning of the ion optics is of particularly important in determining the flight times of ions with respect to initial ion velocity. Modelling of the ion trajectories in a software package such as SIMION (Scientific Instrument Services, Inc.) may provide optimised tuning that further exploits the differences in ion focussing that arises from differences in particle composition. The transferability of the method to other LAAP-ToF instruments will also likely depend on the tuning of the ion optics. Tuning parameters used for this study are provided in the supplement (S1). Establishing the role of fixed instrument design features will indicate how transferable this method is to other designs of single particle mass spectrometers. For example, using extraction by an electric field one would expect an increased plume density and therefore an increase in space-charge effects and collisions as the ion plume is not free to expand in all directions. In addition, an orthogonal geometry of the excimer laser with the particle beam is likely to produce less variation in the position and timing in which a particle encounters the threshold power density for LDI compared to the co-axial geometry used in the LAAP-TOF.

**5 Conclusions**

A novel technique that uses peak centroid measurement in addition to peak areas to describe the mass spectral characteristics arising from the LDI of single particles of silicate mineral dust has been presented. To our knowledge, this is the first time that the properties of a material have been described by the relative changes in the ion arrival times of an ion species at a TOF-MS detector. Examination of the spectral patterns from dust samples reveals spectrum to spectrum variation in the relative peak position of the $SiO_3^-$ molecular ion with respect to the $O^-$ elemental ion that occurs in distinct modes. Comparison of these modes with the borosilicate glass and carbon black suggest that the mode preference is a result of particle crystal structure and elemental composition, the properties that define mineral phase.

Analysis of clay mineral standards and nominally pure feldspars suggest that the relative shift of the elemental and molecular ions is a function of the quantity and co-ordination of potassium and sodium cations in the interstitial complex. It is proposed that the mineral phase of the particle matrix influences the ion formation mechanism and produces variations

in initial ion velocity and ion formation timing during the LDI of single particles. These effects are enhanced by the co-axial geometry of the excimer laser with the particle beam and are preserved in the field free extraction regime in the TOF-MS implemented in the LAAP-TOF. This may represent an important step in the understanding of how LDI proceeds in SPMS.

Analysis of multi-mineralic clay mineral standards reveals a multi-modal pattern in ion arrival times. A scheme that classifies single particles has been defined on the basis of the alkali metal peak areas and the relative difference in the shift in the ion arrival times of the $O^-$ and $SiO_3^-$ species with respect to a calibration, a parameter we call $\tau$. Application of the scheme to clay mineral standards result in the single particle differentiation of illite-smectite clays mineral (ISCM), feldspars and kaolinite that is in agreement with bulk mineralogy reported in semi-quantitative XRD analysis.

The nature of the interstitial complex and its effect on crystal structure can be extremely varied even within a single grain or crystal so that complete reproducibility would not be expected from any single particle measurement of a natural mineral dust sample.  In circumstances in which the actual mineral phase cannot be determined, it is still expected that the ion arrival time ratios will be a useful parameter in describing differences in physiochemical properties of silicate particles . This represents an important step forward in the study of atmospheric processes where single particle mineral phase is important.

**Data Availability**

All laboratory acquired data presented in this manuscript is available by request from the corresponding author.

**Author Contribution**

The experiment was designed and performed by N.Marsden with the support of M.Flynn. The data analysis was developed and performed by N.Marsden. The manuscript was prepared by N.Marsden with contribution from H.Coe and J.Allan.

**Competing Interests**

The authors declare that they have no conflicts of interests.

[revised manuscript text omitted]